# CAPS-1 promotes fusion competence of stationary dense-core vesicles in presynaptic terminals of mammalian neurons

Margherita Farina[1], Rhea van de Bospoort[1], Enqi He[1], Claudia M Persoon[2], Jan RT van Weering[2], Jurjen H Broeke[2], Matthijs Verhage[1,2]*, Ruud F Toonen[1]*

[1]Department of Functional Genomics, Center for Neurogenomics and Cognitive Research, VU University Amsterdam, Amsterdam, Netherlands; [2]Department of Clinical Genetics, VU University Medical Center, Amsterdam, Netherlands

**Abstract** Neuropeptides released from dense-core vesicles (DCVs) modulate neuronal activity, but the molecules driving DCV secretion in mammalian neurons are largely unknown. We studied the role of calcium-activator protein for secretion (CAPS) proteins in neuronal DCV secretion at single vesicle resolution. Endogenous CAPS-1 co-localized with synaptic markers but was not enriched at every synapse. Deletion of CAPS-1 and CAPS-2 did not affect DCV biogenesis, loading, transport or docking, but DCV secretion was reduced by 70% in CAPS-1/CAPS-2 double null mutant (DKO) neurons and remaining fusion events required prolonged stimulation. CAPS deletion specifically reduced secretion of stationary DCVs. CAPS-1-EYFP expression in DKO neurons restored DCV secretion, but CAPS-1-EYFP and DCVs rarely traveled together. Synaptic localization of CAPS-1-EYFP in DKO neurons was calcium dependent and DCV fusion probability correlated with synaptic CAPS-1-EYFP expression. These data indicate that CAPS-1 promotes fusion competence of immobile (tethered) DCVs in presynaptic terminals and that CAPS-1 localization to DCVs is probably not essential for this role.

*For correspondence: matthijs. verhage@cncr.vu.nl (MV); ruud. toonen@cncr.vu.nl (RFT)

**Competing interests:** The authors declare that no competing interests exist.

## Introduction

Neuropeptides and neurotrophic factors are essential for brain development and synaptic plasticity (*McAllister et al., 1996*; *Huang and Reichardt, 2001*; *Poo, 2001*; *Samson and Medcalf, 2006*; *van den Pol, 2012*). These neuromodulators are transported in dense-core vesicles (DCVs). Dysregulation of DCV transport and fusion is associated with cognitive and post-traumatic stress disorders (*Sadakata et al., 2007b*; *Meyer-Lindenberg et al., 2011*; *Sah and Geracioti, 2013*). DCVs bud off at the Golgi network (*Kim et al., 2006*) and are transported via microtubule-based motor proteins (*Hirokawa et al., 2009*; *Schlager and Hoogenraad, 2009*). High-frequency firing facilitates DCV fusion and the resultant calcium influx triggers SNARE complex-dependent DCV secretion (*Bartfai et al., 1988*; *Hartmann et al., 2001*; *de Wit et al., 2009*; *van de Bospoort et al., 2012*). Unlike synaptic vesicles (SVs), DCVs lack a local recycling mechanism. To ensure a constant and uniform supply of DCVs at release sites, DCVs are generally very dynamic although some DCVs are stationary. Stimulation triggers arrest of moving DCVs (*de Wit et al., 2006*; *Shakiryanova et al., 2006*; *Matsuda et al., 2009*; *Wong et al., 2012*), probably promoting their local availability for secretion. DCV fusion sites can be located in the entire neuron but DCVs preferentially fuse at presynaptic terminals and release at extra-synaptic sites requires more robust stimulation (*van de Bospoort et al., 2012*). Recently, we have shown that Munc13 is an important

**eLife digest** Our ability to think and act is due to the remarkable capacity of the brain to process complex information. This involves nerve cells (or neurons) communicating with each other in a rapid and precise manner by releasing synaptic vesicles containing neurotransmitters across the gaps—called synapses—between neurons. In addition to this fast neurotransmitter signalling, neurons can transmit signals by releasing chemical signals called neuropeptides. Neuropeptides are major regulators of human brain function, including mood, anxiety, and social interactions.

Neuropeptides and other neuromodulators such as serotonin and dopamine are normally packaged into bubble-like compartments called dense-core vesicles. Compared to synaptic vesicles we know much less about how dense-core vesicles are trafficked and released. Dense-core vesicles are generally mobile and move around the inside of cells to release neuropeptides where and when they are needed. However, some vesicles are stationary and may even be loosely tethered to the cell membrane. Most of the sites where dense-core vesicles can fuse with the cell membrane are at synapses.

Previous work has suggested that the protein CAPS-1 is important for moving dense-core vesicles to the correct sites on the cell membrane, and for releasing neuropeptides across the synapses of worms and flies. However, detailed insights into this process in mammalian neurons are lacking.

By examining neurons from both normal mice and mice lacking the CAPS-1 protein, Farina et al. have now analyzed the role CAPS-1 plays in releasing neuropeptides. In cells lacking CAPS-1 fewer dense-core vesicles merged with the cell membrane than in cells containing the protein. However, a new technique that tracks the movement of individual vesicles revealed that only stationary dense-core vesicles had difficulties fusing; mobile vesicles continued to fuse with the cell membrane in the normal manner. Introducing CAPS-1 into cells lacking this protein corrected the fusion defect experienced by the stationary vesicles.

Farina et al. also showed that CAPS-1 was present at most—but not all—synapses, and synapses that had more CAPS-1 released more neuropeptides. This work shows that CAPS proteins strongly influence the probability of dense-core vesicle release and that neurons can tune this probability at individual synapses by controlling the expression of CAPS. Future work will be aimed at understanding how neurons can achieve this and which protein domains in CAPS are required.

regulator of DCV fusion at synapses (*van de Bospoort et al., 2012*). However, in contrast to SV release, a comprehensive insight in the molecular mechanisms of DCV secretion is still lacking.

Previous studies in Caenorhabditis *elegans* and *Drosophila* have implicated calcium-activator protein for secretion (CAPS) proteins in DCV secretion. Mutants of the *C. elegans* CAPS ortholog UNC-31 show reduced peptide release without affecting synaptic vesicle fusion (*Speese et al., 2007*). In *C. elegans* neurons, UNC-31 is required for the docking of DCVs at the plasma membrane (*Zhou et al., 2007*; *Hammarlund et al., 2008*; *Lin et al., 2010*). In *Drosophila*, deletion of dCAPS also affects DCV release but in contrast to *C. elegans*, leads to an increased DCV presence in terminals (*Renden et al., 2001*). Mammals express two CAPS genes, CAPS-1 and CAPS-2, which are complementarily expressed in brain (*Speidel et al., 2003*) and are essential for synaptic transmission (*Jockusch et al., 2007*). In adrenal chromaffin cells, CAPS-1 deletion affects catecholamine uptake in chromaffin granules (*Speidel et al., 2005*; *Brunk et al., 2009*) and deletion of CAPS-1 and CAPS-2 abolishes their fusion without affecting docking (*Liu et al., 2010*). CAPS-2 is important for cerebellar development and neuron survival (*Sadakata et al., 2004*, *2007a*), and deletion of CAPS-1 in cerebellar neurons perturbs DCV trafficking (*Sadakata et al., 2010*, *2013*). Hence, these studies suggest that CAPS proteins are involved in several aspects of DCV trafficking and release in invertebrates and in mammalian chromaffin cells, but their role in mammalian versus invertebrate systems appears to differ considerably, especially regarding synaptic transmission and cell survival.

In this study, we analyzed the distribution and function of CAPS proteins in DCV trafficking and fusion in mammalian neurons using wild type (WT), CAPS-2 null mutant and CAPS-1/2 null mutant mice. Endogenous CAPS-1 was present in puncta that partially overlapped with synaptic markers and also co-localized with DCV markers. CAPS deletion did not affect DCV biogenesis, neuropeptide loading or average DCV transport velocity. In CAPS double null mutant (DKO) neurons, DCV fusion

was strongly reduced at synapses and at extra-synaptic sites. We developed a novel release assay to track single DCVs prior to fusion and found that CAPS deletion strongly affected DCV release of stationary, presumably tethered vesicles. We provide evidence that CAPS-1 localization at synapses is calcium dependent and that DCV release probability correlates with synaptic CAPS-1 expression levels.

## Results

### CAPS-1 is present at synapses and overlaps with secretory vesicles markers

To understand CAPS-1 function in neuronal DCV release, we first investigated its sub-cellular localization in cultured neurons. Hippocampal neurons at 14 days in vitro (DIV 14) were stained with a novel, CAPS-1-specific antibody (*Figure 1—figure supplement 1*). CAPS-1 was present in the cytosol and in dendritic and axonal puncta, probably membrane domains (*Figure 1A*). Approximately 45% of these CAPS-1 puncta co-localized with the synaptic marker VGLUT1 in the entire neuron (*Figure 1B, C*, Pearson's coefficient: 0.42±0.04, n=7). CAPS-1 immunoreactivity was detectable in approximately 60% of VGLUT1 positive synapses and vice versa approximately 60% of the CAPS-1 puncta co-localized with VGLUT1 (*Figure 1B, D*, Manders' coefficients CAPS-1 in VGLUT1: 0.67±0.08, and VGLUT1 in CAPS-1: 0.64±0.05, n=8). CAPS-1 domains were also found at extra-synaptic sites (*Figure 1B, E*, white arrowheads). The CAPS-1 expression pattern differed from the sub-cellular localization of the DCV priming protein Munc13-1, which was much more restricted to the synapse (*Figure 1E, F*, Pearson's coefficient in the entire neuron: 0.64±0.04, n=10) with 96% of synapses containing M13-1 (*Figure 1G*, number of synapses containing M13-1: 96.3±0.7%, n=4, number of synapses containing CAPS-1: 58.1±10.9%, n=7). Since CAPS proteins have been initially identified as DCV resident proteins (*Berwin et al., 1998*), we tested the co-localization of CAPS-1 with DCVs in hippocampal neurons. Using the endogenous DCV protein chromogranin B (ChrB) we found that the majority of DCVs are located in the axon (*Figure 1—figure supplement 2*). Antibody incompatibility precluded co-staining of CAPS-1 antibody with antibodies against ChrB. Therefore, we used the DCV cargo neuropeptide Y (NPY) fused to Venus (*Nagai et al., 2002*), which showed more than 80% co-localization with this endogenous marker (*Figure 1H, I*, Manders' coefficients for chromogranin B in NPY-Venus puncta: 0.97±0.02, and NPY-Venus in ChrB puncta: 0.84±0.01, n=14). Approximately 35% of NPY-Venus labeled DCVs co-localized with endogenous CAPS-1 (*Figure 1J–L*, Pearson's coefficient: 0.45±0.05, n=21, number of NPY-labeled DCVs co-expressing CAPS-1: 34.71±3.03%, n=6). These data show that endogenous CAPS-1 is present at many but not all synapses. In addition, CAPS-1 domains are found at extra-synaptic regions and CAPS-1 partly co-localizes with DCV markers.

### DCV secretion is severely reduced upon CAPS deletion in hippocampal neurons

CAPS proteins are implicated in catecholamine uptake into secretory vesicles in chromaffin cells (*Speidel et al., 2005*). As a defect in vesicle loading could influence our analysis of release events, we first analyzed possible effects of CAPS deletion on the loading of proteins into DCVs. The fluorescence intensity distribution of single DCV release events measured with the DCV cargo Semaphorin-3A coupled to pH-sensitve GFP (SemapHluorin, see below) in WT and CAPS DKO cells was similar; indicating that SemapHluorin loading of fusing DCVs was not affected in CAPS DKO neurons (*Figure 2A*). To analyze SemapHluorin loading of all DCVs in the cell, we quantified the fluorescence intensity of SemapHluorin labeled DCVs upon application of $NH_4^+$ (which instantly de-quenches all intra-vesicular pHluorin, *Figure 2B*) and of antibody-labeled endogenous DCV cargo protein, secretogranin II, in WT and CAPS DKO neurons. $NH_4^+$ application showed that loading of SemapHluorin was comparable between WT and CAPS DKO neurons (*Figure 2C, D*). Also, fluorescence intensity levels of secretogranin II were unchanged between WT and CAPS DKO neurons (*Figure 2E*). As neuronal viability is affected in CAPS-2 null mutant cerebellum (*Sadakata et al., 2004*, *2007a*), we analyzed neuronal morphology and DCV numbers in CAPS DKO neurons and did not find differences between WT and DKO neurons (*Figure 2F–H*). Hence, neuronal morphology, DCV biogenesis and protein loading in hippocampal neurons are not affected by deletion of CAPS-1 and CAPS-2.

To examine the function of CAPS proteins in DCV secretion we used two different fluorescent DCV cargo proteins in two secretion assays, hippocampal mass cultures and isolated single neuron cultures. First, DCVs were labeled with the secreted axon-guidance protein Semaphorin 3A coupled to the

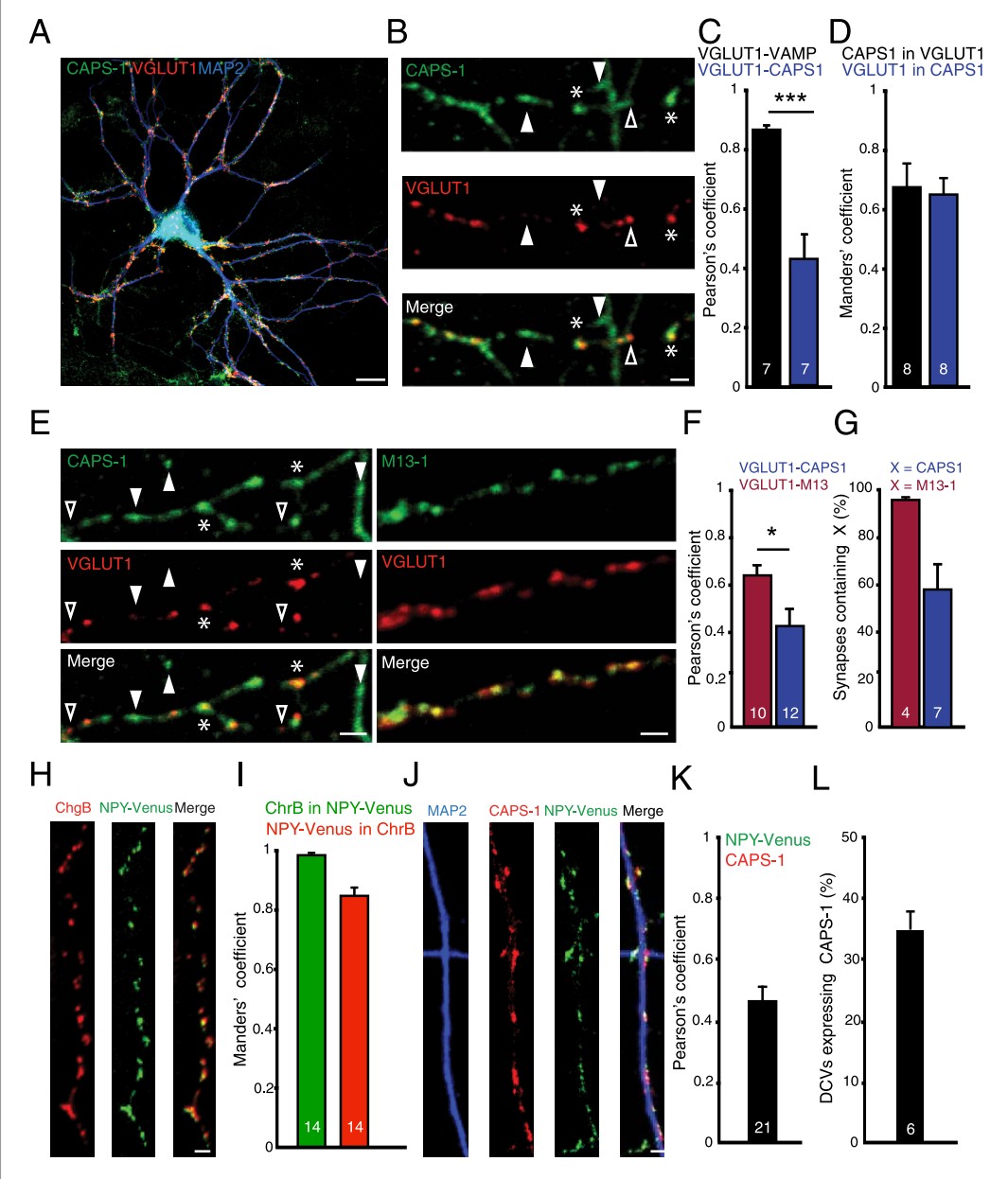

**Figure 1**. CAPS-1 clusters are present at synaptic and extra-synaptic sites and partly co-localize with DCVs.
(**A**) Example image of a hippocampal neuron (DIV 14) stained for endogenous CAPS-1 (green), dendrite marker
MAP2 (blue) and synapse marker VGLUT1 (red). Scale bar 10 μm. (**B**) Zoom of a neurite stained for CAPS-1 and
VGLUT1. CAPS-1 rich domains not overlapping with VGLUT1 (filled arrowhead), VGLUT1 punctum not enriched for
CAPS-1 (open arrowhead), VGLUT1 puncta overlapping with a CAPS-1 rich domain (stars). Scale bar 2 μm. (**C**) Co-
localization of CAPS-1 with VGLUT1 in the entire neuron quantified by Pearson's correlation. Co-localization of
VAMP2 with VGLUT1 was used as a positive control (VGLUT1-VAMP2: 0.8±0.02, n=7 neurons; VGLUT1-CAPS-1: 0.4±
0.05, n=7 neurons, ***p<0.0001). (**D**) Mander's coefficients for the proportion of CAPS-1 immuno-reactivity in
VGLUT1 positive locations: 0.67±0.08, n=8 neurons or proportion of VGLUT1 immunoreactivity in CAPS-1 positive
locations: 0.64±0.05, n=8 neurons. (**E**) Example images of neurites from hippocampal neurons (DIV 14) stained for
endogenous CAPS-1 (green, left panel) and VGLUT1 (red) or for Munc13-1 (M13-1, green, right panel) and VGLUT1
(red). CAPS-1 domains not overlapping with VGLUT1 (filled arrowheads), VGLUT1 puncta not enriched for CAPS-1
(open arrowheads). Synapses (VGLUT1 puncta) overlapping with CAPS-1 rich domains (stars). Scale bar 5 μm.
(**F**) CAPS-1 co-localization with VGLUT1 in the entire neuron is lower compared to co-localization of Munc13-1 and
VGLUT1. Pearson's correlation M13-1-VGLUT1: 0.6±0.04, n=10; CAPS-1-VGLUT1: 0.4±0.08, n=12, *p<0.05.
(**G**) Percentage of VGLUT1 labeled synapses expressing CAPS-1 is lower than VGLUT1 labeled synapses expressing

*Figure 1. continued on next page*

*Figure 1. Continued*

(VGLUT1/CAPS-1: 58.5±10.9%, n=7 neurons, number of synapses = 239; VGLUT1/M13-1: 96.2±0.7%, n=4 neurons, number of synapses = 350). (**H**) Example images of neurites from hippocampal neurons (DIV 14) infected with lentivirus encoding NPY-Venus (green) and stained for chromogranin B (ChgB, red). Scale bar 2 μm. (**I**) Mander's coefficients for the proportion of endogenous ChrB immuno-reactivity in NPY-Venus puncta: 0.97±0.02, n=14 neurons or proportion of NPY-Venus immunoreactivity in ChrB puncta: 0.84±0.01, n=14 neurons. (**J**) Example images of neurites from hippocampal neurons (DIV 14) infected with lentivirus encoding NPY-Venus and stained for CAPS-1 (red) and MAP2 (blue). Scale bar 2 μm. (**K**) Quantification of co-localization of CAPS-1 and NPY-Venus in the entire neuron. Pearson's coefficient: 0.45±0.05; n=21 neurons. (**L**) Percentage of NPY-Venus labeled DCVs co-localizing with CAPS-1: 34.71±3.03%, n=6, number of DCVs = 414.

The following figure supplements are available for figure 1:

**Figure supplement 1**. Specificity of CAPS-1 antibody.

**Figure supplement 2**. Endogenous DCV marker (chromogranin **B**) distribution in isolated single neurons.

pH-sensitive enhanced green fluorescent protein (EGFP) variant, pHluorin (SemapHluorin, *Figure 3A, B*) and release was measured in CAPS-1/CAPS-2 DKO (*Jockusch et al., 2007*) and WT hippocampal neurons in DIV 14 mass cultures (*Figure 3A*). We have previously shown that SemapHluorin labels all DCVs in cultured neurons and reports single DCV fusion events as a sudden increase in fluorescence upon opening of the fusion pore (*de Wit et al., 2009*; *van de Bospoort et al., 2012*). DCV release was triggered by electrical stimulation using a protocol known to elicit robust neuropeptide release from neurons (16 bursts of 50 action potentials at 50 Hz, *Hartmann et al., 2001*; *de Wit et al., 2009*; *van de Bospoort et al., 2012*). *Figure 3B* top panel shows a typical fusion event reported by SemapHluorin: upon fusion pore opening intravesicular pH rises sharply unquenching pHluorin, which results in a strong increase in fluorescence. Fluorescence increase of two standard deviations above initial fluorescence (*Figure 3B*, grey dotted line) was scored as fusion event in panels C and E. After this increase, fluorescence may remain high (left trace, persistent event) or may decay (middle and right trace, transient events). Both persistent and transient events were counted as fusion events in panels C and E. *Figure 3—figure supplement 1* explains the different fusion modes reported by SemapHluorin in more detail (see also *de Wit et al., 2009*).

Upon stimulation, CAPS DKO neurons showed a more than 60% reduction in the number of DCV fusion events compared to WT (*Figure 3C*, WT: 23.7±4.5 events/field of view, n=24; CAPS DKO: 9.6±1.5 events/field of view, n=32, N=4, p<0.01). The average vesicle fusion rate during stimulation, calculated from the cumulative release plots (*Figure 3D*), was 1.2±0.4 vesicles/s for WT compared to 0.15±0.1 vesicles/s for CAPS DKO neurons. Also when we zoomed in on the first four bursts of 50 action potentials, the number of fusion events was significantly lower in CAPS DKO neurons compared to WT (*Figure 3E, F*). Together, these findings show that deletion of both CAPS isoforms strongly reduces activity-dependent secretion of neuronal DCVs by decreasing the total number of fusing vesicles and vesicle release rate during stimulation.

## CAPS-1 is the CAPS isoform responsible for loss of DCV secretion in hippocampal neurons

Mass cultures can be used to assess the average number of release events per field of view, not per neuron. Therefore, we adapted a single neuron culture protocol using micro-islands of astrocytes (*Bekkers and Stevens, 1991*; *Wierda et al., 2007*), which enables imaging of the entire axonal and dendritic arbor of a single neuron to perform quantitative single-cell DCV release measurements (*Figure 4A*). DCVs were labeled with NPY-pHluorin, and vesicle fusion was triggered by electrical stimulation (as in *Figure 3*). As CAPS-2 expression in hippocampal neurons is very low and deletion of CAPS-2 does not affect SV release (*Jockusch et al., 2007*), we used neurons from CAPS-2 null mutant (CAPS-2KO) littermates as controls. The number of DCV fusion events per cell reported by the sudden increase in fluorescence upon fusion pore opening was strongly decreased in CAPS DKO compared to controls (*Figure 4B, C*, CAPS-2KO: 49.6±16.3 events/cell, n=9; CAPS DKO: 10.6±7.9 events/cell, n=5, N=3; p<0.05). Hence, CAPS-1 deletion strongly inhibits DCV fusion in isolated hippocampal neurons.

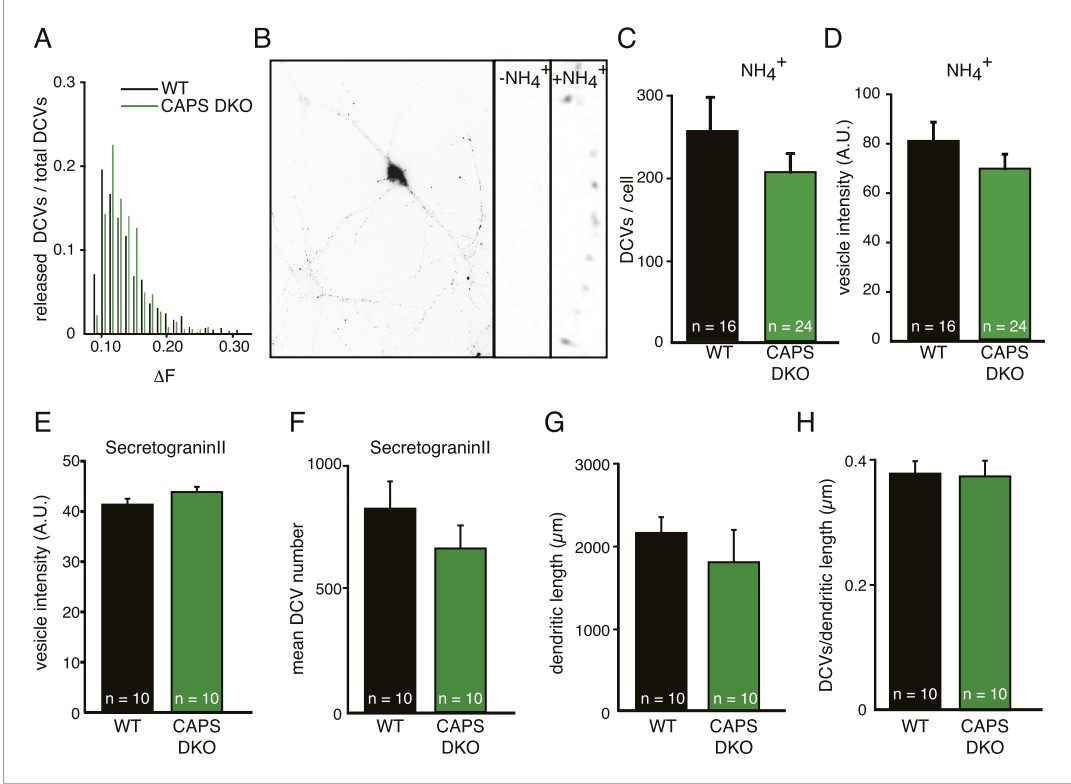

**Figure 2**. CAPS deletion does not influence DCV peptide loading or DCV biogenesis in hippocampal neurons.
(**A**) Frequency distribution of fluorescence intensity increase ($\Delta F$) of individual DCV fusion events in WT and CAPS DKO neurons. No major differences are observed in $\Delta F$ of fusion events between WT and CAPS DKO neurons indicating similar Semaphluorin loading per DCV in WT and CAPS DKO neurons. The number of DCVs per bin is normalized to the total number of DCVs released. (**B**) Inverted wide-field image of a neuron expressing Semaphluorin upon $NH_4^+$ application to reveal all DCVs present in the cell. Zoom shows the effect of vesicle de-acidification upon $NH_4^+$ application ($-NH_4^+$ before application, $+NH4^+$ during application). (**C**) Average number of DCV puncta per neuron quantified from the $NH_4^+$ response is similar in WT (n=16 neurons) and CAPS DKO (n=24) neurons. (**D**) Average intensity (in arbitrary units, AU) of single DCV puncta quantified from the $NH_4^+$ response in WT and CAPS DKO neurons is similar (WT n=16 neurons and 4435 puncta, CAPS DKO n=24 neurons and 4892 puncta). (**E**) Average intensity (in AU) of single DCV puncta in the field of view of confocal images is similar in non-transfected WT (n=10 neurons) and CAPS DKO (n=10) neurons stained for the endogenous DCV cargo secretogranin II and the dendritic marker MAP2. (**F**) Average number of DCV puncta per field of view. (**G**) Average dendritic length per field of view. (**H**) Number of DCV puncta per dendritic length.

DCV fusion pore opening can progress to complete release of cargo or to partial release and fusion pore resealing (*Alabi and Tsien, 2013*; *Wu et al., 2014*). To investigate these release modes in CAPS-2 KO and CAPS DKO neurons, we labeled DCVs with NPY-mCherry, which reports full cargo release events as complete disappearance of fluorescent puncta (*Figure 4D*). The average number of such fusion events in control cells was much lower compared to events reported by NPY-pHluorin, which reports all fusion events irrespective of complete or incomplete release of cargo (compare *Figure 4E* and *Figure 4B*, NPY-mCherry CAPS-2KO: 14.6±3.2 events/cell; NPY-pHluorin CAPS-2KO: 49.6±16.3 events/cell). However, also with this reporter a more than 50% decrease in the number of fusion events per cell was observed in CAPS DKO neurons compared to CAPS-2KO (*Figure 4E, F*, CAPS-2KO: 14.6±3.2 events/cell, n=14; CAPS DKO: 3.4±1.3 events/cell, n=8, N=3, p<0.01). In addition to the strong decrease in DCV fusion events in CAPS DKO neurons, in almost 50% of CAPS DKO cells expressing either NPY-pHluorin or NPY-mCherry electrical stimulation failed to induce vesicle fusion (*Figure 4G*, these cells were excluded from analysis in *Figure 4B–F*).

We previously showed that DCVs preferentially fuse at synapses and that Munc13 plays a crucial role in synaptic preference of DCV release (*van de Bospoort et al., 2012*). To test whether CAPS

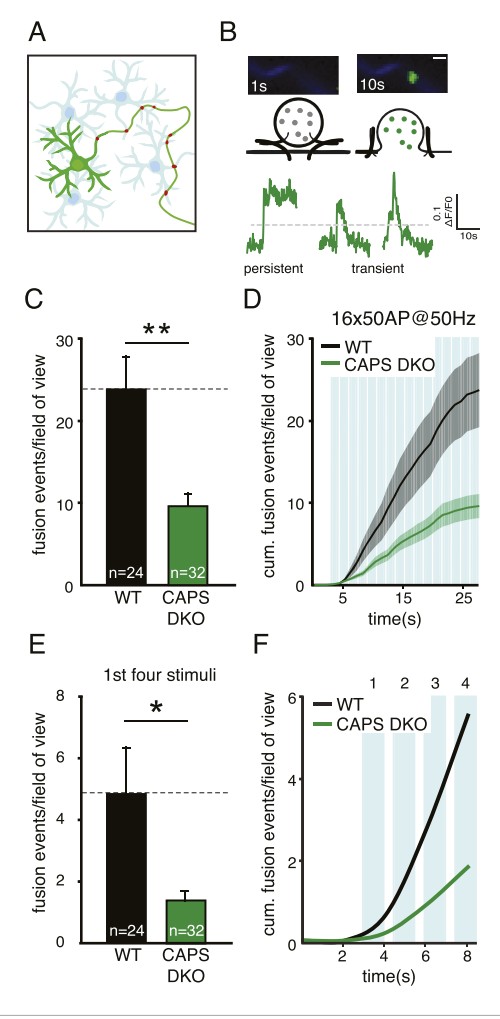

**Figure 3**. CAPS deletion reduces DCV fusion in hippocampal neurons. (**A**) Schematic drawing of a neuronal mass culture used in these experiments in which one neuron expresses a fluorescently tagged morphology marker (green) and a DCV marker (red). Non-labeled neurons are indicated in light blue. (**B**) Top panel: two stills showing a typical example of a fusion event of a SemapHluorin labeled DCV with the corresponding schematic drawing. Fusion pore opening causes a sudden increase of fluorescence intensity corresponding to the increase in intravesicular pH. Bottom panel: example traces of SemapHluorin labeled DCV fusion events. Fluorescence increase of two standard deviations above initial fluorescence (grey dotted line) was scored as fusion event in **C** and **E**. After which, fluorescence may decrease (transient events) or remain high (persistent events). *Figure 3—figure supplement 1* explains this behavior in detail. Scale bar 1 µm. (**C**) Average number of DCV fusion events per field of view during electrical stimulation with 16 bursts of 50 AP at 50 Hz with 0.5 s interval (WT: 23.7±4.5, n=24 neurons; CAPS DKO: 9.6±1.5, n=32 neurons, N=4, independent experiments, **p<0.01). (**D**) Cumulative frequency plot of DCV fusion events during stimulation

deletion affected synaptic preference of DCV fusion we labeled presynaptic terminals with the live cell marker synapsin-mCherry and analyzed NPY-pHluorin fusion events (*Figure 4H*). Unlike Munc13 deletion, deletion of CAPS affected extra-synaptic and synaptic DCV release to the same extent (*Figure 4I*).

To confirm that the secretion defect in CAPS DKO neurons was due to lack of CAPS-1, CAPS-1 was re-introduced in CAPS DKO cells at 10 DIV via lenti-virus infection ('rescue') and DCV release was tested 4 days later. CAPS-1 expression levels in rescued neurons were comparable to endogenous levels (*Figure 4J*). Introduction of CAPS-1 in CAPS DKO cells rescued the secretion defect (*Figure 4K*) and release kinetics were similar to control cells (*Figure 4L*). Thus, deletion of CAPS-1 resulted in a strong reduction of DCV fusion events reported by NPY-PHluorin or NPY-mCherry and fusion at synapses and extra-synaptic sites was affected to the same extent.

## CAPS-1 deletion strongly affects fusion of stationary DCVs

DCVs can be highly mobile or stationary (*de Wit et al., 2006*; *Wong et al., 2012*; *Goodwin and Juo, 2013*). Both stationary and mobile vesicles fused upon electrical stimulation (*Figure 5A*). In control neurons the majority of DCV fusion events were stationary DCVs (*Figure 5B*, CAPS-2KO: stationary 12.2±2.8 fusion events per cell, moving 4.0±1.0 fusion events per cell, n=10, N=3, p<0.05). However, in CAPS DKO neurons, release of stationary vesicles was strongly decreased while release of mobile vesicles was similar to CAPS-2KO (*Figure 5B*, CAPS DKO: stationary 2.0±0.8 fusion events per cell, moving 3.0±1.1 events per cell, n=5, N=2).

To test if CAPS deletion affected general DCV trafficking, we analyzed the dynamics of DCVs that did not fuse during the 90s imaging protocol. No differences between CAPS-2KO and CAPS DKO neurons were observed for these vesicles (*Figure 5C*, CAPS-2KO: stationary 33.8±1.2, moving: 17.6±0.7, n=5, p<0.05; CAPS DKO: stationary 33.66±3.0, moving 19.33±2.4, n=3, p<0.05). We also did not detect differences in average velocity of fusing and non-fusing DCVs prior to stimulation, during stimulation or in the period after stimulation (*Figure 5D, E*).

CAPS/UNC-31 deletion reduces the number of docked DCVs in *C. elegans* neuromuscular junctions (*Hammarlund et al., 2008*) while the total number of DCVs in mammalian synapses is not different between control and CAPS DKO (*Jockusch et al., 2007*). We examined electron

*Figure 3. Continued*

(average vesicle fusion rate per cell during stimulation WT: 1.2±0.4 vesicles/s; CAPS-1/2 DKO: 0.15±0.1 vesicles/s). Blue bars represent stimulation of 16×50 AP at 50 Hz with 0.5 s interval. (**E**) Number of DCV fusion events per field of view during the first four bursts of the stimulation in **C**, (WT: 4.8±1.5, n=24; CAPS DKO: 1.4± 0.3, n=32, *p<0.05). (**F**) Cumulative frequency plot of DCV fusion events during the first four bursts of the stimulation, showing that the initial release rate is slower in CAPS DKO neurons.

The following figure supplement is available for figure 3:

**Figure supplement 1**. Different fusion events reported by SemapHluorin.

micrographs of synapses from CAPS DKO and control neurons to test if DCV localization was affected by deletion of CAPS. No morphological differences between the two groups were observed (*Figure 5F* and *Figure 5—figure supplement 1D*). The number of synapses containing DCVs (*Figure 5G*), the percentage of docked DCVs (*Figure 5H*, see *Figure 5—figure supplement 1A–C* for our definition of docked vesicles), the distance of DCVs to the closest plasma membrane (*Figure 5I*), and the average number of DCVs per synapse (*Figure 5J, K*) were similar between CAPS DKO and control synapses. Thus, deletion of CAPS-1 does not affect transport of DCVs or DCV localization at synapses but specifically affects fusing vesicles, by reducing the number of fusion-competent stationary vesicles.

## CAPS-1-EYFP fusion protein replaces endogenous CAPS-1 in DCV secretion and synaptic transmission

To investigate the intracellular dynamics of CAPS-1, we introduced CAPS-1 fused to enhanced yellow fluorescent protein (EYFP) (CAPS-1-EYFP) in CAPS DKO neurons (*Figure 6A*). As for endogenous CAPS-1 (*Figure 1*), the majority of CAPS-1-EYFP co-localized with the live-synaptic marker synapsin-ECFP (*Figure 6A*, bottom panel, stars). In addition, we observed, again similar to endogenous CAPS-1, synapses without detectable CAPS-1-EYFP (*Figure 6A*, bottom panel, open arrowhead) and CAPS-1 rich domains that did not co-localize with synapses (*Figure 6A*, bottom panel, filled arrowhead).

CAPS-1-EYFP efficiently rescued DCV secretion in CAPS DKO neurons (*Figure 6B*) and also rescued synaptic transmission. Whole cell patch-clamp recordings of single neurons on micro-dot astrocyte islands revealed no differences between control and CAPS-1-EYFP rescued CAPS DKO neurons. Evoked postsynaptic current amplitude (EPSC, *Figure 6C, D*) and spontaneous release characteristics (mEPSC frequency and amplitude, *Figure 6E–G*) were similar to controls. Also EPSC rundown during high-frequency stimulation (40 Hz) and its recovery (*Figure 6H, I*) were completely rescued. Hence, CAPS-1-EYFP mimics endogenous CAPS-1 and was used for further experiments to study the dynamics of this protein in living cells.

## Localization of CAPS-1 at synapses is calcium dependent

We studied the dynamics of CAPS-1-EYFP upon electrical stimulation using the same stimulation paradigm used to elicit DCV release (16×50 AP at 50 Hz). We infected hippocampal neurons with lentiviral particles encoding CAPS-1-EYFP and synapsin-ECFP and imaged the two fluorophores simultaneously (*Figure 7A, B*). During stimulation, the fluorescence intensity of CAPS-1-EYFP and synapsin-ECFP at synapses strongly decreased while the extra-synaptic intensities of both proteins increased (*Figure 7A–F*). A large variability of intensity changes upon stimulation was observed between cells. In some neurons fluorescence intensity of synapsin-ECFP and CAPS-1-EYFP returned to baseline within 3 min (*Figure 7C, D*), while in other cells the intensities of both proteins remained below their initial fluorescence (*Figure 7E, F*). Fluorescence intensities of membrane-bound EYFP did not change at synapses during stimulation showing that the decrease of CAPS-1-EYFP fluorescence at synapses reported dispersion of the protein into neurites (*Figure 7D, F*, insets). The average response of synaptic CAPS-1-EYFP showed a similar dynamic profile as synapsin-ECFP, dispersing from synapses during and up until 3 min after calcium influx, while extra-synaptic CAPS-1-EYFP fluorescence increased during stimulation (*Figure 7G, H*). Thus, during stimulation a fraction of synaptic CAPS-1 redistributes from synapses into neurites similar to the synaptic vesicle protein synapsin 1.

## DCV release probability correlates with CAPS-1 expression at single synapses

We showed that on a cellular level, removing CAPS-1 strongly impairs DCV fusion (*Figure 3* and *Figure 4*). To test whether at the single synapse level CAPS expression levels correlated with DCV

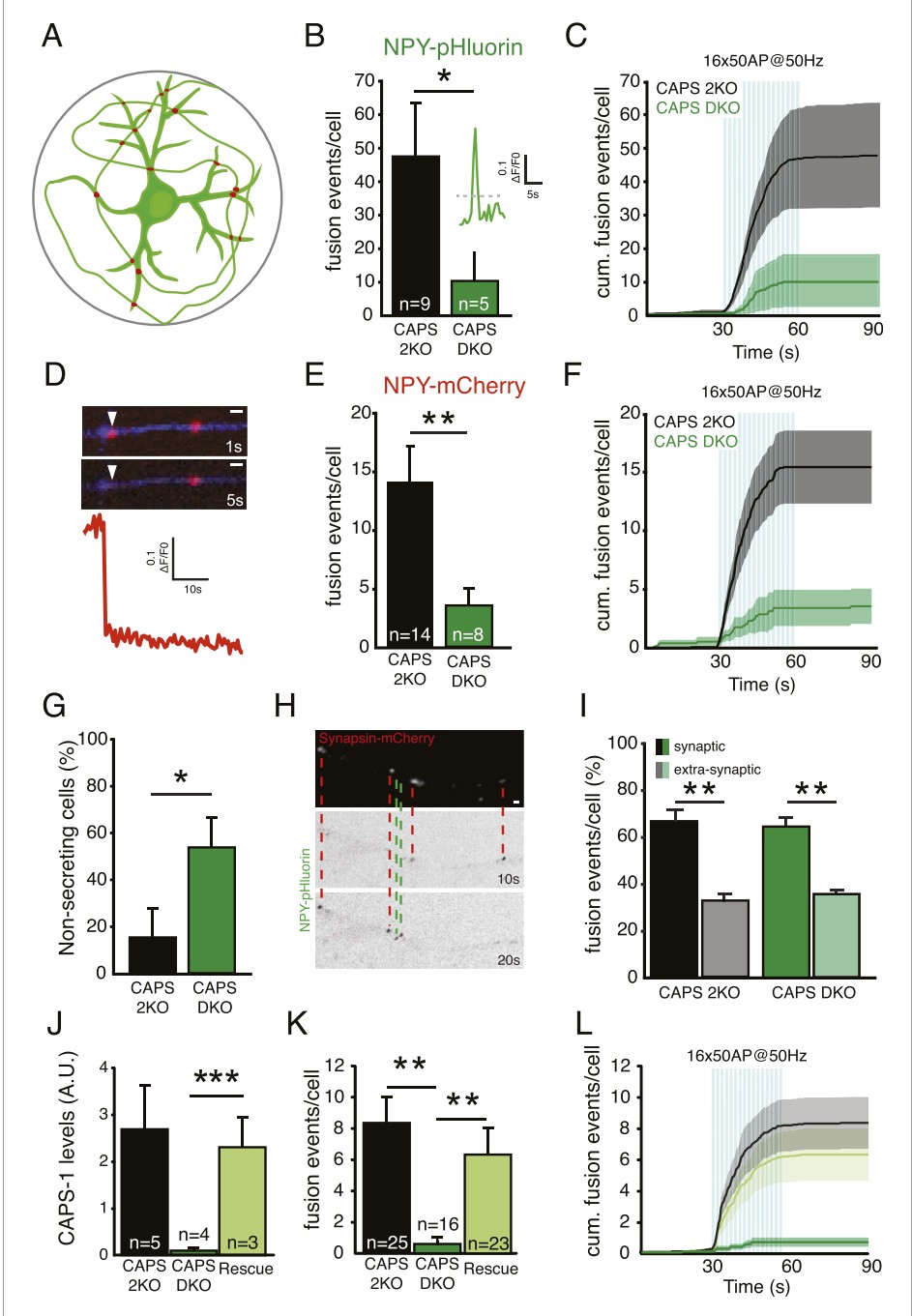

**Figure 4**. CAPS-1 deletion impairs DCV fusion in isolated neurons. (**A**) Schematic drawing of a single isolated neuron grown on a micro island of astrocytes and expressing fluorescently tagged morphology marker (green) and a synapse marker (red) in addition to either NPY-pHluorin or NPY-mCherry (not shown). Average island diameter is 375 μm. (**B**) Average number of DCV fusion events per cell upon electrical stimulation of 16 bursts of 50 AP at 50 Hz using NPY-pHluorin as DCV marker (CAPS-2KO: 49.6±16.3, n=9; CAPS DKO: 10.6±7.9, n=5, number of independent experiments (N)=3, *p<0.05). Inset shows a typical example of a fusion event reported by NPY-pHluorin. Fluorescence increase of two standard deviations above initial fluorescence (grey dotted lines) was scored as fusion event in **B**. (**C**) Cumulative frequency plot of DCV fusion events in **B**, showing that DCV release is triggered by electrical stimulation paradigm (blue bars represent 16 bursts of 50 AP at 50 Hz, 16×50 AP at 50 Hz). (**D**) White arrowhead: Typical example of a fusion event with complete cargo release reported by the sudden and complete disappearance of NPY-mCherry fluorescence intensity. These fusion events were in measured in **E** and **K**. Time in seconds (s) after start of stimulation. Neurite marker is ECFP (blue). Scale bar 1 μm. (**E**) Average number of fusion

*Figure 4. continued on next page*

*Figure 4. Continued*

events with complete cargo release per cell upon electrical stimulation using NPY-mCherry as DCV marker (CAPS-2KO: 14.6±3.3, n=14; CAPS DKO: 3.4±1.4, n=8, N=3, **p<0.01). (**F**) Cumulative frequency plot of DCV fusion events with complete cargo release in **E**, (blue bars represent 16 bursts of 50 AP at 50 Hz, 16×50 AP at 50 Hz).
(**G**) Percentage of non-secreting cells is increased in CAPS DKO neurons (CAPS-2KO: 15±12.4%, n=26; CAPS DKO 53.6±12.6%, n=26, *p<0.05). Non-secreting cells were excluded from the analyses in **B** and **E**. (**H**) Example images showing NPY-pHluorin labeled DCVs fusing at synapsin-mCherry labeled synapses, indicated by the red dashed lines and DCV fusion events at extra-synaptic sites, indicated by the green dashed lines. Bar is 1 μm. (**I**) Percentage of synaptic and extra-synaptic DCV release events measured with NPY-pHluorin shows a similar distribution in CAPS DKO compared to CAPS-2KO (CAPS-2KO synaptic: 67.0±3.6, CAPS-2KO extra-synaptic: 0.32±1.4, n=20, **p<0.01; CAPS DKO synaptic: 64.7±2.3, CAPS DKO extrasynaptic: 35.6±0.9, n=28, **p<0.01). (**J**) CAPS-1 expression levels assessed by semi-quantitative immunofluorescence in CAPS-2KO, CAPS DKO, and CAPS DKO expressing CAPS-1 (Rescue) (CAPS-2KO: 2.8±0.7 AU, n=5; CAPS DKO: 0.1±0.0, n=4; Rescue: 2.4±0.7, n=3, ***p<0.0001). (**K**) Average number of fusion events leading to complete release per cell upon electrical stimulation with NPY-mCherry as DCV marker in CAPS-2KO, CAPS DKO and Rescue (CAPS-2KO: 8.5±1.7, n=25; CAPS DKO: 0.7±0.3, n=16; Rescue: 6.4±1.7, n=23, N=6). (**L**) Cumulative frequency plot of complete release DCV fusion events in **K**, (blue bars represent 16 bursts of 50 AP at 50 Hz, 16×50 AP at 50 Hz).

release probability we simultaneously imaged NPY-mCherry labeled DCVs and CAPS-1-EYFP (*Figure 8A*). Like endogenous CAPS-1, CAPS-1-EYFP was present in puncta (*Figure 8A*, left top panel). The majority of these were stationary throughout the imaging experiment (*Figure 8A, B*). Only 5% of all CAPS-1-EYFP puncta were mobile during our imaging paradigm (*Figure 8B*). Furthermore, the number of mobile CAPS-1-EYFP puncta that co-trafficked with dynamic NPY-mCherry labeled DCVs was very low (seven out of 106 mobile DCVs showed co-trafficking of CAPS-1-EYFP, *Figure 8C*).

As CAPS-1 co-localizes with synaptic markers but is not enriched at every synapse (*Figure 1A–C*, *Figure 8D*), we tested whether CAPS-1 positive synapses (*Figure 8D*, star) have a higher DCV release probability than CAPS-1-EYFP deficient synapses (*Figure 8D*, open arrowhead). Synapses expressing CAPS-1-EFYP secreted more DCVs upon stimulation than synapses with no detectable CAPS-1-EYFP expression (*Figure 8E*, percentage of DCVs released at CAPS-1-EYFP positive synapses: 84.0±3.3%, −CAPS-1: 16.0±3.1%, n=83 DCVs from five cells, ***p<0.0001). Further-more, DCV fusion in synapses with no detectable CAPS-1-EYFP expression required more prolonged stimulation than in CAPS-1-EYFP positive synapses (*Figure 8F*). Also, CAPS-1 enriched synapses released more DCVs from a stationary pool than from a mobile pool compared to CAPS-1 deficient synapses (*Figure 8G*, CAPS-1-EYFP positive synapses: stationary DCVs: 46; moving DCVs: 23; total DCVs counted: 69; CAPS-1-EYFP negative synapses: stationary DCVs: 6; moving DCVs: 8; DCVs counted: 14). Together these results show that DCVs preferentially reside and fuse at CAPS-1 rich domains and that CAPS-1-EYFP and DCVs rarely travel together in DIV 14 neurons. Furthermore, DCV release probability correlated with synaptic CAPS-1 expression levels.

## Discussion

Here we report that, in mammalian neurons, CAPS-1 is an important regulator of DCV fusion. CAPS-1 is present at synapses and co-localizes with DCV markers but rarely travels together with dynamic DCVs. Using two DCV secretion assays we found that CAPS-1 deletion strongly impaired synaptic and extra-synaptic DCV release in hippocampal neurons. CAPS-1 deletion did not reduce the presence or docking of DCVs at synaptic sites, but strongly affected DCV secretion of stationary DCVs. Synaptic expression levels of CAPS-1 were modulated by neuronal activity and correlated with DCV release probability. CAPS-1 thus functions as a priming protein at synaptic and extra-synaptic sites, promoting the fusion competence of stationary DCVs.

### CAPS-1 is present at mammalian synapses and overlaps with secretory vesicles markers

CAPS-1 was identified as a brain protein (*Walent et al., 1992*) that binds membranes associated with DCV secretion (*Berwin et al., 1998*) and stimulates calcium-dependent fusion of secretory vesicles in PC12 cells (*Mennerick et al., 1995*; *Loyet et al., 1998*; *Grishanin et al., 2002*). This suggests that CAPS-1 might be a DCV resident protein that is delivered to sites of exocytosis on DCVs. However, we

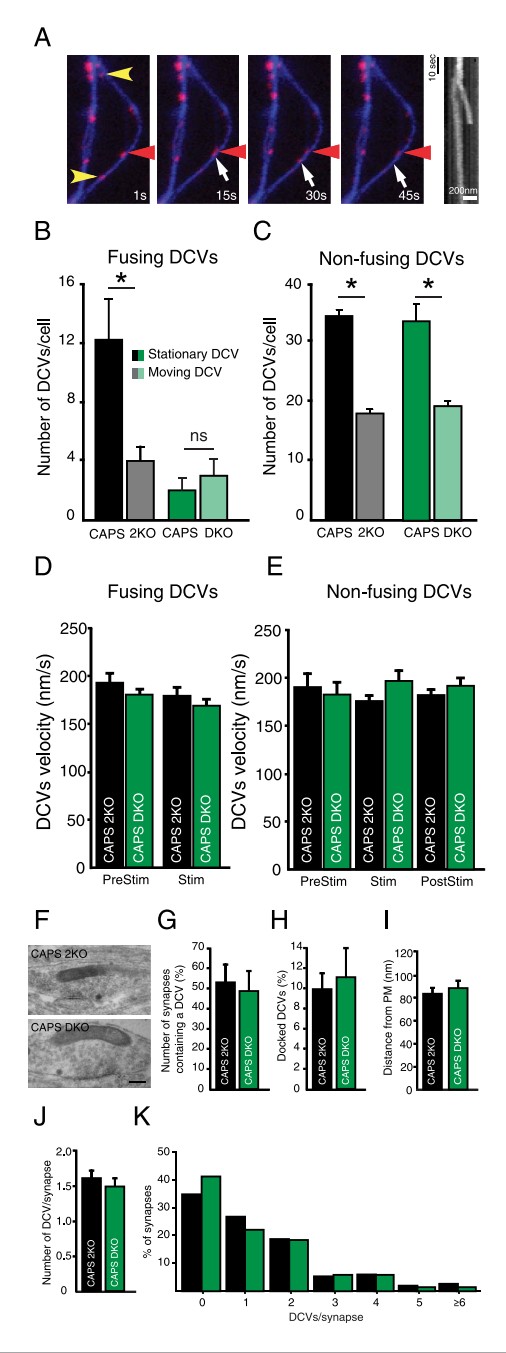

**Figure 5**. Deletion of CAPS-1 affects fusion of stationary DCVs. (**A**) Stills from DCV fusion assay. Electrical stimulation starts at second 10. Yellow arrowheads indicate stationary vesicles that fuse, red arrowhead indicates a stationary vesicle that does not fuse and white arrow shows a moving DCV that fuses. Kymograph shows the trajectories of the stationary and moving vesicle over time. (**B**) Fusing DCVs classified as stationary or moving showing that CAPS deletion strongly affects fusion from stationary vesicles (CAPS-2KO stationary prior to fusion: 12.2±2.8, moving: 4±1.0, n=10, *p<0.05; CAPS DKO stationary: 2.3±1.3, moving: 4.0±1.7, n=3, ns, N=3). (**C**) Non-fusing DCVs classified as
*Figure 5. continued on next page*

observed that in mammalian neurons endogenous CAPS-1 localized in axonal and dendritic domains that often, but not always, co-localized with synaptic and DCV markers. Cell-wide CAPS-1 co-localization with the canonical DCV marker NPY yielded an average co-localization of only 35%. Furthermore, in CAPS DKO neurons rescued with CAPS-1-EYFP, DCVs and CAPS-1-EYFP only rarely traveled together. Finally, the increased dynamics of synaptic CAPS-1-EYFP with CAPS-1 redistribution upon stimulation are best explained by diffusion (see below) but not with a redistribution of vesicular CAPS-1. Thus, in mature mammalian neurons, CAPS-1 does not appear to be a general DCV-resident protein, but a dynamic, cytosolic factor translocating to synapses and extra-synaptic sites in an activity-dependent manner to promote fusion of pre-docked (tethered) DCVs.

## A post-docking role for CAPS-1 in DCV fusion

*Figure 3* and *Figure 4* show that in the absence of CAPS-1, DCV fusion in hippocampal neurons is severely impaired, without defects in vesicle biogenesis and loading of endogenous or exogenous DCV cargo proteins. In addition, analysis of DCV dynamics in our live cell imaging experiments and intra-synaptic localization on electron micrographs of CAPS DKO neurons revealed that CAPS-1 deletion does not affect DCV trafficking along microtubules, synaptic accumulation and membrane docking of DCVs. Together, this suggests that in mature neurons most CAPS-1 molecules interact with DCVs after these vesicles arrived at the plasma membrane. This conclusion is consistent with observations in *Drosophila* neuromuscular junctions where deletion of dCaps results in an increase in the number of DCVs present at synapses (*Renden et al., 2001*) and the lack of a docking defect upon deletion of CAPS-1/-2 in adrenal chromaffin cells (*Liu et al., 2010*). However, chemical fixation used in our study might produce artifacts that change the precise distance between vesicles and the membrane. As a consequence, vesicles touching the membrane in fixed tissue may in fact have been at a short distance from the membrane. Recent excellent work using cryo-fixation and EM tomography to analyse docking of SVs and DCVs in central synapses of several mutant mouse strains including CAPS DKO showed that SV docking is impaired in CAPS DKO neurons. In line with our findings, DCVs were present in similar numbers in CAPS DKO synapses. However, although not statistically significant, the distribution of DCVs

*Figure 5. Continued*

stationary or moving showing that CAPS deletion does not affect general trafficking behavior of non-fusing vesicles (CAPS-2KO stationary: 33.8±1.2, moving: 17.6± 0.7, n=5, *p<0.05; CAPS DKO stationary: 33.7±3.0, moving: 19.3±2.4, n=3, *p<0.05). (**D**) Average velocity of DCVs classified as moving prior to fusion in **B**, before stimulation (PreStim = 30 s; CAPS-2KO: 193.6±9.9 nm/s, n=169 vesicles, CAPS DKO: 180.4±9.2 nm/s, n=18) and during electrical stimulation (Stim: from second 30 to the onset of fusion; CAPS-2KO = 180.9±6.6 nm/s, n=169; CAPS DKO: 169.2±7.5, n=18). (**E**) Average velocity of non-fusing DCVs in **C**, before stimulation (PreStim = 30 s; CAPS-2KO: 190.8±13.7 nm/s, n=168, CAPS DKO 186.8±13.5 nm/s, n=91), during stimulation (Stim = 24 s; CAPS-2KO: 175.3±6.7 nm/s, n=168; CAPS DKO 201.8±11.8 nm/s, n=91) and after stimulation (PostStim = 36 s CAPS-2KO: 182.2±6.2 nm/s, n=168, CAPS DKO 196.2±8.5 nm/s, n=91). (**F**) Typical examples of electron micrographs of neuronal DCVs in synapses. Scale bar 50 nm. (**G**) Number of synapses containing one or more DCVs (CAPS-2KO: 53.9±12.1%, n=198 synapses; CAPS DKO 47.6±5.4%, n=152 synapses, N=4). (**H**) Percentage of docked DCVs per synapse (CAPS-2KO: 10.3±1.7%, DCVs = 285, CAPS DKO: 11.1±3.7%, DCVs = 201). (**I**) Average distance of DCVs to the closest plasma membrane (CAPS-2KO: 81.2±4.8; CAPS DKO: 84.9±10.4). (**J**) Average number of DCVs per synapse (CAPS-2KO: 1.6±0.2; CAPS DKO: 1.5±0.2). (**K**) Frequency distribution of number of DCVs per synapse (% synapses, normalized to the total number of DCVs per group).

The following figure supplement is available for figure 5:

**Figure supplement 1**. DCV docking definition and zooms of *Figure 5F*.

within 200 nm of the active zone appeared to be reduced (*Imig et al., 2014*). Hence, general consensus exists on a post-synapse delivery role for CAPS but future studies using cryo-fixation might unmask (subtle) docking defects upon CAPS loss.

CAPS-1 likely functions at the final stages of DCV fusion, interacting with proteins and lipids that function in DCV docking, priming and fusion (*Grishanin et al., 2004*; *Sadakata et al., 2007b*; *James et al., 2009*; *Parsaud et al., 2013*; *Sah and Geracioti, 2013*). CAPS-1 changes the conformation of syntaxin-1 from a state incompatible with SNARE-complex formation ('closed-state'), to a state that allows formation of functional SNARE complexes ('open-state'), as release in CAPS DKO chromaffin cells is rescued by expression of an 'open'-variant of syntaxin-1 (*Liu et al., 2010*) and overexpression of open syntaxin can bypass the requirement for CAPS in DCV docking in *C. elegans* (*Hammarlund et al., 2008*). CAPS-1 shares this characteristic with Munc13-1 (*Augustin et al., 1999*; *van de Bospoort et al., 2012*). CAPS and Munc13 proteins may operate in the same molecular priming pathway (*Richmond et al., 2001*; *Jockusch et al., 2007*; *Zhou et al., 2007*) in a non-redundant manner. The fact that both proteins appear to bind syntaxin-1 via different binding modes may account for this non-redundancy (*Parsaud et al., 2013*). A recent paper indeed showed that CAPS-2 and Munc13 use different mechanisms to prime vesicles, whereas Munc13-dependent priming requires its MUN domain this domain in CAPS-2 is dispensable for priming. Instead CAPS-2 appears to require its PIP2 binding pleckstrin homology

domain (*Nguyen Truong et al., 2014*). We found that both proteins play important stimulatory roles in DCV release from mammalian neurons. However, functional differences were evident: CAPS-1 deletion resulted in a larger reduction of DCV release events compared to deletion of Munc13-1 with a ±70% reduction in CAPS DKO (this study) and ±60% reduction in Munc13 DKO neurons (*van de Bospoort et al., 2012*). For synaptic vesicle fusion the situation is opposite: Munc13 DKO neurons clearly show larger defects in synaptic transmission than CAPS DKO neurons (*Varoqueaux et al., 2002*; *Jockusch et al., 2007*).

CAPS-1 displayed a different expression pattern than Munc13-1. Munc13-1 expression is strictly synaptic and specifically supports DCV secretion from synapses, while being dispensable for extra-synaptic secretion (*van de Bospoort et al., 2012*). In contrast, CAPS-1 accumulations are found at many synapses, but not all, and at extra-synaptic sites along neurites. Also, CAPS-1 deletion equally affects DCV release from synapses and from extra-synaptic sites. Finally, CAPS-1 redistributes from synapses during activity while Munc13-1 does not (*Kalla et al., 2006*). These differences in (re) distribution suggest that (1) CAPS-1 domains at extra-synaptic sites promote DCV release independently of Munc13-1. However, CAPS-1-dependent extra-synaptic release is less efficient than synaptic DCV release, which indicates that the concerted action of CAPS-1 and Munc13-1 is most efficient in priming DCVs for release. (2) At synapses, CAPS-1 and Munc13-1 have non-redundant roles in promoting DCV release, similar to their non-redundancy in SV release (*Jockusch et al., 2007*). The larger effect of CAPS deletion on DCV release may be explained by the additional reduction of extra-synaptic release events compared to Munc13-1 deletion. Finally, some extra-synaptic DCV

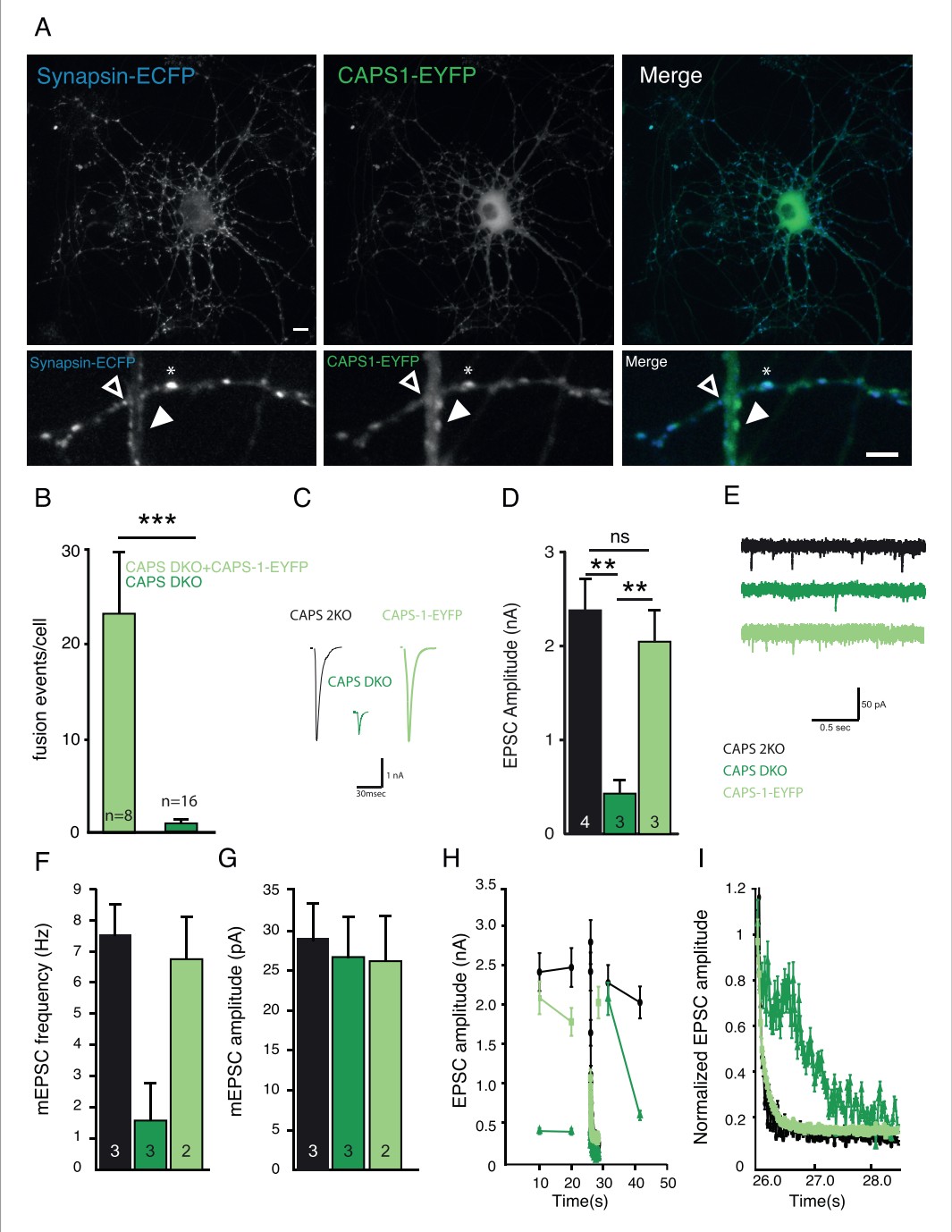

**Figure 6**. CAPS-1-EYFP fusion protein replaces CAPS-1 in DCV secretion and synaptic transmission. (**A**) Isolated single CAPS DKO neuron grown on glia micro-island expressing CAPS-1-EYFP (green) and synapsin-ECFP (blue). Top panels: maximum projection of the entire neuron, Scale bar 10 μm. Bottom panels: zooms of top panels showing synapses without detectable CAPS-1-EYFP (open arrowhead), CAPS-1 rich domain not overlapping with synapsin-ECFP (filled arrowhead) and a CAPS-1 rich synapse (star). Scale bar 5 μm. (**B**) Average number of DCV fusion events per cell (events/cell) upon electrical stimulation (16×50 AP at 50 Hz) using NPY-pHluorin as DCV marker in CAPS DKO neurons with (+CAPS-1-EYFP) and without (CAPS DKO) CAPS-1-EYFP (+CAPS-1-EYFP: 23.0±6.8, n=8, CAPS DKO: 0.7±0.3, n=16, ***p<0.0001, Mann–Whitney test). (**C**) Example traces of evoked EPSCs in CAPS-2KO (black), CAPS DKO (dark green) and CAPS-1-EYFP (light green) rescued CAPS DKO neurons. (**D**) Expression of CAPS-EYFP rescues EPSC amplitude in CAPS DKO neurons (CAPS-2KO: 2.4±0.3 nA, n=4; CAPS DKO: 0.48±0.15 nA, n=3; CAPS-1-EYFP: 2.0±0.4 nA, n=3, **p<0.01). (**E**) Example traces of spontaneous release (mEPSCs). (**F**) Mean

*Figure 6. continued on next page*

*Figure 6. Continued*

mEPSC frequency. (CAPS-2KO: 7.5±1.1 Hz, n=3; CAPS DKO: 1.7±1.0 Hz, n=3; CAPS-1-EYFP: 6.9±1.3 Hz, n=2). (**G**) Mean mEPSC amplitude. (CAPS-2KO: 29.0±4.1 pA, n=3; CAPS DKO: 27.1±5.0 pA, n=3; CAPS-1-EYFP: 26.8±7.2 pA, n=2). (**H**) Changes in EPSC amplitude induced by 100-pulse train at 40 Hz during low frequency (0.1 Hz) stimulation. The interval between low- and high-frequency stimulation is 3 s (**I**) 100-pulses at 40 Hz induced rundown of normalized EPSC amplitude (zoom of **H**).

release remained in CAPS DKO neurons indicating that extra-synaptic DCV release can occur in the absence of CAPS and Munc13 albeit with very low release probability.

## Synaptic activity reorganizes synaptic CAPS-1 levels

Strong stimulation triggered redistribution of synaptic CAPS-1 into the axonal shaft. This redistribution was very similar to synapsin-ECFP (our work and *Richmond et al., 2001*; *Parsaud et al., 2013*) and is also reported for other synaptic proteins like Rab3a (*Tsuriel et al., 2009*), Syntaxin-1 (*Ribrault et al., 2011*) and Munc18 (*Cijsouw et al., 2014*) and for SVs (*Cheung and Cousin, 2011*). In contrast, the mobility of active zone proteins like Munc13-1 and Bassoon is not affected by acute stimulation (*Kalla et al., 2006*; *Tsuriel et al., 2009*). Hence, synapses rapidly exchange part of their components during high frequency stimulation. This allows synapses to adapt their release probability during and directly after stimulation as we showed for Munc18-1 (*Cijsouw et al., 2014*). This is also an attractive explanation for the re-distribution of CAPS-1 as we found that synapses with increased CAPS-1 expression levels had a higher release probability than synapses with low/no CAPS-1. CAPS-1 binds to PI 4,5-P2 (PIP2) (*Loyet et al., 1998*) and localizes to PIP2 clusters in the plasma membrane via its PIP2-binding pleckstrin homology (PH) domain (*James et al., 2008*). The PH domain is also required to prime secretory vesicles (*Kabachinski et al., 2014*; *Nguyen Truong et al., 2014*). Robust $Ca^{2+}$ influx in our experiments likely activates phospholipase C, which hydrolysis PIP2 and may trigger CAPS-1 dispersion from synapses. Hence, calcium-dependent PIP2 hydrolysis may act as negative feedback mechanism reducing CAPS-1 availability at the synapse after robust stimulation. In addition to direct membrane interaction, CAPS-1 also binds syntaxin via its MUN domain. Syntaxin also disperses from synapses resulting in CAPS-1 co-dispersion. As Munc13-1 does not show activity-dependent redistribution among synapses but instead increases its membrane-bound fraction upon calcium influx, synapses appear to utilize two distinct mechanisms to control release probabilities during/after high frequency stimulation.

## Materials and methods

### Plasmids

SemapHluorin was generated by replacing EGFP in Sema3A-EGFP (*De Wit et al., 2005*) with super-ecliptic pHluorin (SpH) (*de Wit et al., 2009*). NPY-Venus was previously described (*Nagai et al., 2002*) and NPY-SpH was generated by replacing Venus with SpH (*de Wit et al., 2009*). Synapsin-mCherry was a kind gift of Dr A Jeromin (Allen Brain Institute, Seattle, USA) and synapsin-ECFP was obtained by replacing mCherry with ECFP. CAPS-1 (KIAA1121-Kazusa DNA) was sequence verified and cloned as CAPS-1-ires-EGFP and EYFP-CAPS-1. All constructs but SemapHluorin were subcloned into pLenti vectors that were produced as described (*Naldini et al., 1996*). Transduction efficiencies were tested on HEK cells.

### Laboratory animals

CAPS-1/2 double knockout mice have been described before (*Jockusch et al., 2007*). Mouse embryos were obtained by caesarean section of pregnant females from timed mating. Animals were housed and bred according to institutional, Dutch and US governmental guidelines.

### Primary neuron cultures

Dissociated hippocampal neurons were prepared from embryonic day 18 mice as described (*de Wit et al., 2009*). Hippocampi were dissected in Hanks buffered salts solution (HBSS, Sigma, The Netherlands) and digested with 0.25% trypsin (Invitrogen, The Netherlands) for 20 min at 37°C.

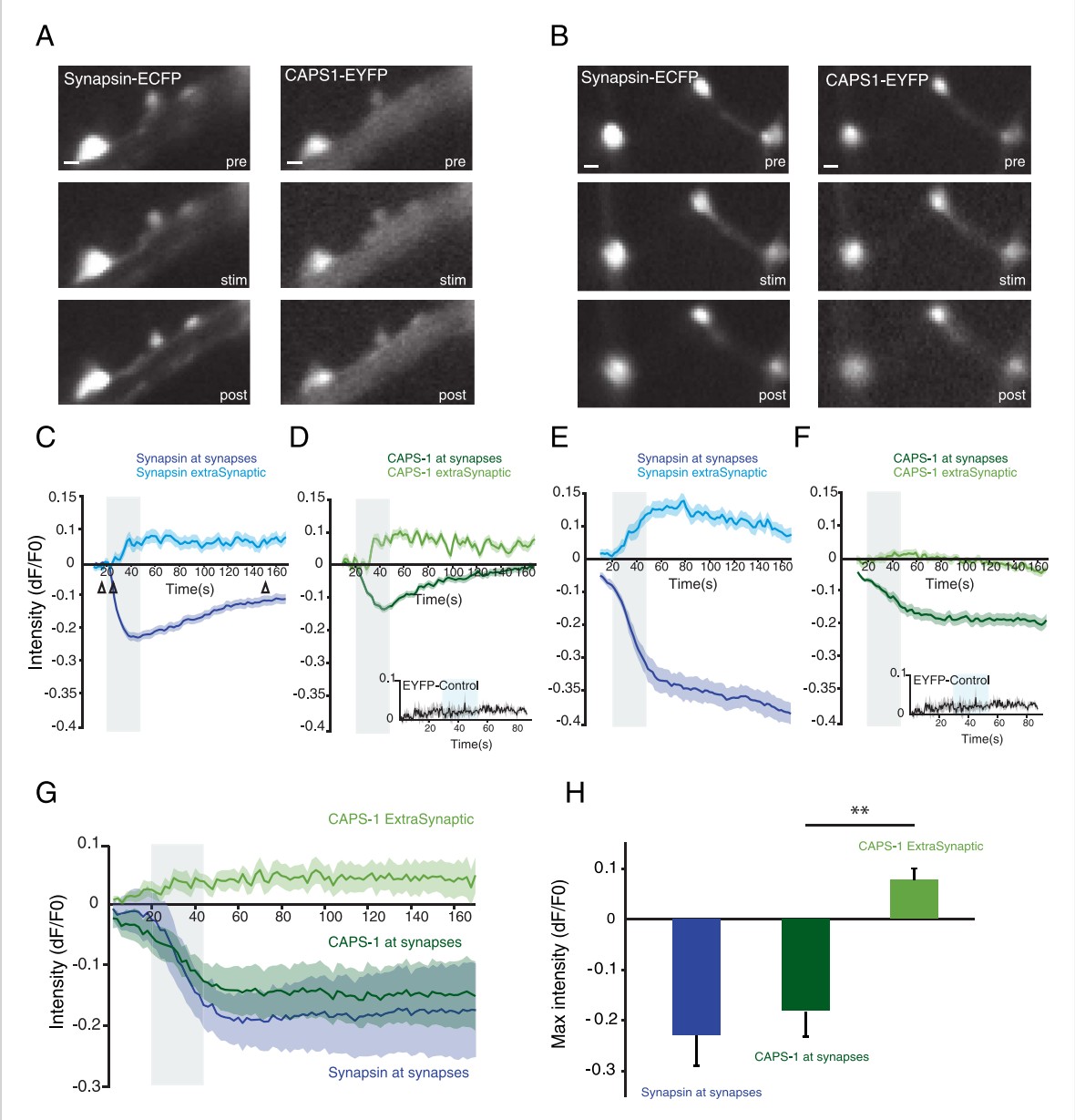

**Figure 7**. Localization of CAPS-1 at synapses is calcium dependent. (**A** and **B**) Grey scale images of synapsin-ECFP (left) and CAPS-1-EYFP (right) labeled synapses showing the same region before (pre-15s) during (stim-22s) and after (post-150s) electrical stimulation (16×50 AP at 50 Hz). Synapsin-ECFP and CAPS-1-EYFP were imaged simultaneously at 0.5 Hz. (**C** and **E**) Traces of relative intensity changes ($\Delta F/F_0$) of synapsin-ECFP at synapses and extra-synaptic locations (79 synaptic and 31 extra-synaptic) showing increased extra-synaptic and decreased synaptic fluorescence upon stimulation (16×50 AP at 50 Hz). Open arrowheads in **C** indicate the pre- and post stimulations time points (same for **D**–**F**). (**D** and **F**) Example traces of relative intensity changes ($\Delta F/F_0$) of CAPS-1-EYFP at synapsin-ECFP labeled synapses from **C**. showing increased extra-synaptic and decreased synaptic fluorescence upon stimulation (16×50 AP at 50 Hz). Inset: synaptic fluorescence of membrane associated EYFP (EYFP control) as control. (**G**) Average relative intensity profiles of synapsin-ECFP, and CAPS-1-EYFP at synapses and CAPS-1-EYFP extra-synaptic, (395 synaptic regions, 155 extrasynaptic regions, n=5 cells). (**H**) Maximum relative intensity changes (max $\Delta F/F_0$) of synapsin-ECFP, and CAPS-1 at synapses and CAPS-1 at extra synaptic regions at t=160 s calculated from G (**p<0.01, n=5 cells each).

Hippocampi were washed and triturated with fire-polished Pasteur pipettes, counted and plated in neurobasal medium (Invitrogen) supplemented with 2% B-27 (Invitrogen), 1.8% HEPES, 1% glutamax (Invitrogen) and 1% Pen-Strep (Invitrogen). High-density cultures (25,000 neurons/well) were seeded on pre-grown cultures of rat glia cells (37,500 cells/well) on 18 mm glass coverslips in 12-well plates. For

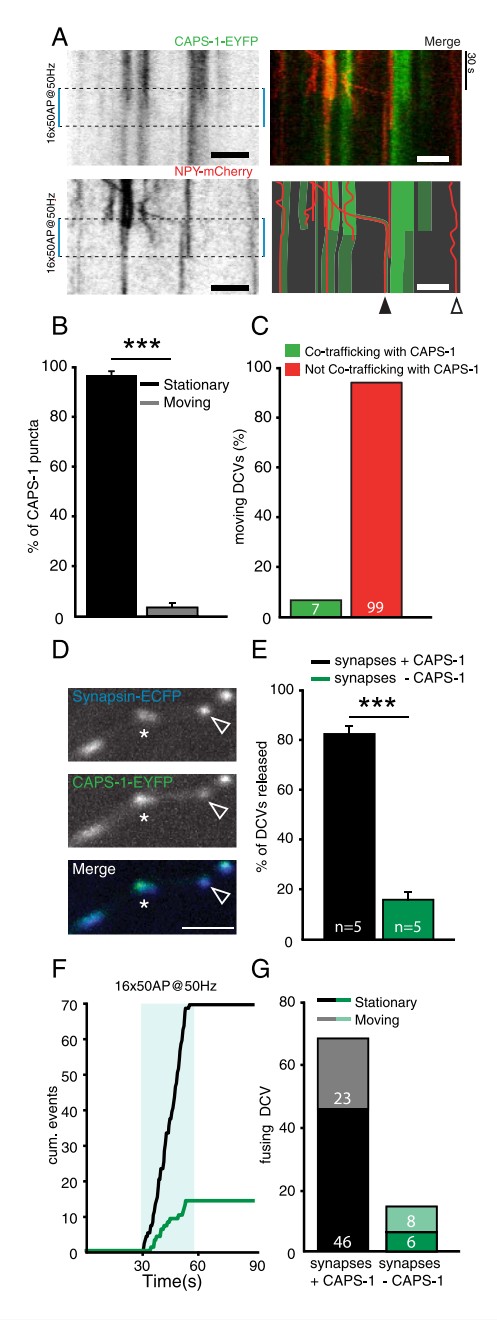

**Figure 8**. Presence of CAPS-1 increases DCV release probability at single synapses. (**A**) Left top and bottom panels: kymographs of CAPS-1-EYFP and NPY-mCherry imaged simultaneously (acquisition frequency 2 Hz) for 90 s and stimulated at second 30 (dashed box, 16×50 AP at 50 Hz). Note the dispersion of the majority of CAPS-1-EYFP puncta upon stimulation and the disappearance of NPY-mCherry puncta upon stimulation. Top right panel shows the merge of the two channels. The bottom right panel shows a schematic drawing of the merged CAPS-1-EYFP and NPY-mCherry channels to aid in the interpretation of the kymographs. Open arrowhead indicates DCV not co-localizing with CAPS-1-EYFP. Filled arrowhead indicates a CAPS-1-EYFP punctum

*Figure 8. continued on next page*

micro-island culture originally described by (*Mennerick et al., 1995*), hippocampal neurons were plated at a density of 2000 neurons/well of a 12-well plate on micro-islands of rat glia as in *Wierda et al., 2007*. These micro-islands were generated by plating 8000/well rat glia on UV-sterilized agarose-coated etched glass coverslips stamped with a 0.1 mg/ml poly-D-lysine (Sigma) and 0.2 mg/ml rat tail collagen (BD Biosciences, The Netherlands) solution.

## Infection and transfection

At 10 DIV, neuronal cultures were infected with a combination of lentiviruses encoding NPY-pHluorin, NPY-mCherry, synapsin-mCherry, CAPS-ires-EGFP, CAPS-YFP or synapsin-ECFP. Alternatively, neurons were transfected using calcium phosphate and expression plasmids for SemapHluorin and synapsin-mCherry. Neurons were imaged at DIV 14–DIV 18.

## Imaging

Coverslips were placed in an imaging chamber perfused with Tyrode's solution (2 mM CaCl$_2$, 2.5 mM KCl, 119 mM NaCl, 2 mM MgCl$_2$, 20 mM glucose and 25 mM HEPES, pH 7.4). All live imaging experiments were performed on a custom-built tandem illumination microscope (TIM; Olympus, The Netherlands) consisting of an inverted imaging microscope (IX81; Olympus) and an upright laser-scanning microscope. The inverted microscope part was used for imaging fluorescence using an MT20 light source (Olympus), appropriate filter sets (Semrock, Rochester, NY), and a 40× oil objective (NA 1.3), or 60× (NA 1.49) for experiments in *Figure 7*, on an EM charge-coupled device camera (C9100-02; Hamamatsu Photonics, Japan). Xcellence RT imaging software (Olympus) was used to control the microscope and record the images.

In pHluorin experiments intracellular pH was neutralized with Tyrode's solution containing 50 mM ammonium chloride (NH$_4$Cl), which replaced sodium chloride (NaCl) on an equimolar basis. Ammonium ion (NH$_4^+$) solution was delivered by gravity flow through a capillary placed onto the cells. To stimulate the cells electrically, parallel platinum electrodes placed close to the cell soma delivered 30 mA, 1 ms pulses controlled by a Master 8 system (AMPI, Germany) and a stimulus generator (A385RC, World Precision Instruments, Germany). The stimulus used was 16 trains of 50 action potentials at 50 Hz with 0.5 s interval. All imaging experiments were performed at room temperature (RT; 21–24°C). For DCV fusion assays imaging frequency used was

*Figure 8. Continued*

co-trafficking with a mobile DCV. Scale bar 5 μm. (**B**) Percentage of stationary and moving CAPS-1-EYFP puncta during image acquisition as described in (**A**). (Stationary: 96.6±1.7, moving: 3.3±1.7, number of cells = 9; number of kymographs per cell = 3, total number of puncta = 116). (**C**) Percentage of mobile DCVs co-trafficking with CAPS-1-EYFP (mobile DCVs not co-trafficking with CAPS-1-EYFP (green bar): 99 or 93.4%; mobile DCVs co-trafficking with CAPS-1-EYFP: 7 or 6.3%, total number of cells = 20, moving DCVs analyzed = 106). (**D**) Typical examples of synapsin-ECFP labeled synapses with high expression levels of CAPS-1EYFP (star) or low expression levels of CAPS-1-EYFP (open arrowhead). (**E**) Percentage of DCV release events occurring at synapses enriched for CAPS-1 (black bar) or depleted for CAPS-1 (green bar), (synapses + CAPS-1: 84.0±3.3%, synapses − CAPS-1: 16.0±3.3%, total DCVs released = 84, n=5, ***p<0.0001). (**F**) Cumulative frequency plot of the DCV release events in **C**. showing that release at CAPS-1 deficient synapses is reduced and delayed. Blue box represents 16×50 AP at 50 Hz stimulation. (**G**) Released DCVs categorized in stationary or moving before fusion (synapses + CAPS-1: stationary DCVs = 46, moving DCVs = 23, total DCVs released = 69; synapse − CAPS-1: stationary DCVs = 6 moving DCVs = 8, total DCVs released = 14).

2 Hz. For SemapHluorin experiments in continental cultures fields of view were selected for presence of SemapHluorin-positive somata, which were placed in the center of the field of view. For protein dispersion experiments, CAPS-EYFP and Synapsin-ECFP were imaged at 0.5 Hz simultaneously for 3 min and stimulated with field electrodes (16×50 AP at 50 Hz). In *Figure 8*, NPY-mCherry and CAPS-EYFP were imaged simultaneously at 2 Hz.

## Image analysis

Stacks from time-lapse recordings acquired with 0.5 s intervals were used to analyze DCV release. A 2×2 pixel region (0.4×0.4 μm) was analyzed according to the experiment as follows. Sema and NPY-pHluorin: fluorescent traces were expressed as fluorescence change ($\Delta F$) compared to initial fluorescence ($F_0$), obtained by averaging the first four frames of the time-lapse recording. A fusion event was counted when fluorescence showed a sudden increase two standard deviations above $F_0$. Onset of fusion was defined as the first frame with an increase of fluorescence of two standard deviations above $F_0$. A cargo-pHl release event or punctum was scored as synaptic when the fluorescence center of such a release event/punctum was within 200 nm (±1 pixel, the approximate minimal point spread function of our system) of the Synapsin-mCherry fluorescence centroid. Extra-synaptic events were all events that did not meet this criterion. We only measured release events from neurites and excluded somatic release events. Somatic release events cannot be reliably measured using wide-field fluorescence microcopy due to the bright fluorescence from vesicles in/near the Golgi apparatus in which the intraluminal pH is not yet acidic. The total number of vesicles was automatically analyzed from the $NH_4^+$ application time lapse using SynD software (*Schmitz et al., 2011*). When using NPY-mCherry: only the fusion events were scored in which NPY-mCherry fluorescence completely disappeared from a 2×2 pixel punctum after bleaching correction (ImageJ Bleaching correction plug-in). DCVs were categorized as stationary or moving based on the slope of the Kymopgraph (ImageJ, MultipleKymograph), if the slope of the line over the kymograph was different from 0 at any point of the movie, the DCV was considered moving.

CAPS-1-ires-EGFP was used to rescue CAPS DKO neurons in combination with NPY-mCherry for DCV fusion assays in *Figure 4*.

Protein dispersion was analyzed by placing regions of interest (ROIs) at the synapses and analyzing the $\Delta F$ over $F_0$ (average of the first four frames) of Synapsin-ECFP and CAPS-1-EYFP over time. ROIs not overlapping with Synapsin-ECFP were chosen for analyzing CAPS-1-EYFP dispersion at extra-synaptic sites ($\Delta F$ as above). These analyses were performed after bleaching correction ($\Delta F$ of the soma over time was used as bleaching and subtracted to the measurements). Membrane associated myristoylated EYFP was used as negative control. For co-localization analysis we used ImagJ software (National Institute of Health, USA, Plug-in JACoP). Pearson's coefficients were calculated to obtain cell wide correlation of fluorescent intensities and Mander's coefficients to obtain co-occurrence in VGLUT positive synapses, CAPS-1 puncta or NPY-puncta (*Figure 1*).

## Electrophysiological recordings

Electrophysiological recordings were performed on single isolated glutamatergic hippocampal neurons between 14 and 18 DIV at RT (21–24˚C). The patch-pipette was filled with a solution containing 135 mM potassium gluconate, 10 mM HEPES, 1 mM ethylene glycol tetra acetic acid (EGTA), 4.6 mM magnesium chloride (MgCl$_2$), 4 mM sodium-Adenosine 5′-triphosphate (Na-ATP), 15 mM creatine phosphate, 50 U/ml phosphocreatine kinase, and 300 milliosmole (mOsm), pH 7.3. The standard

extracellular medium consisted of 140 mM NaCl, 2.4 mM potassium chloride (KCl), 10 mM HEPES, 10 mM glucose, 4 mM calcium chloride (CaCl$_2$), 4 mM MgCl$_2$, and 300 mOsm, pH 7.3. Recordings were performed with an Axopatch 200A amplifier (Molecular Devices, Sunnyvale, CA). Digidata 1322A and Clampex 9.0 (Molecular Devices) were used for signal acquisition. After whole-cell mode, only cells with access resistance of <12 MΩ and leak current of <500 pA were accepted for analysis. Pipette resistance ranged from 4 to 6 MΩ. EPSCs were evoked by depolarizing the cell from −70 to +30 mV for 0.5 ms.

### Fixation and immunocytochemistry

Cells were fixed in 4% formaldehyde (Electron Microscopies Sciences, Germany) in phosphate-buffered saline (PBS), pH 7.4, for 20 min at RT and washed in PBS. First cells were permeabilized for 5 min in PBS containing 0.5% Triton X-100 (Sigma–Aldrich) then incubated for 30 min with PBS (Gibco, The Netherlands) containing 2% normal goat serum and 0.1% Triton X-100. Incubations with primary and secondary antibodies were done for 1–2 hr at RT. Primary antibodies used were: polyclonal MAP2 (Abcam, United Kingdom, 1:500), monoclonal VAMP2 (SySy, Germany, 1:2000) and polyclonal Munc13 (SySy, 1:1000), polyclonal chromogranin B (SySy, 1:500), VGLUT1 (SySy, 1:5000), CAPS-1 (SySy,1:200), and polyclonal secretogranin II (kind gift from P Rosa, Institute of Neuroscience, Milan, Italy). Alexa Fluor conjugated secondary antibodies were from Invitrogen. Coverslips were mounted in Mowiol and examined on a Zeiss LSM 510 confocal laser-scanning microscope with a 40× objective (NA 1.3) or 60× (NA 1.4).

### Electron microscopy

Neurons were fixed at DIV 14 for 1–2 hr at RT with 0.1 M cacodylate buffer, 0.25 mM CaCl$_2$, 0.5 mM MgCl$_2$ (pH 7.4) and processed as described (*Wierda et al., 2007*). Cells were washed three times for 5 min with 0.1 M cacodylate buffer (pH 7.4), post-fixed for 2 hr at RT with 1% osmium tetroxide/1% potassium ferro-cyanide, washed and stained with 1% uranyl acetate for 40 min in the dark. Cells were dehydrated with a series of increasing ethanol concentration steps and embedded in Epon and polymerized for 24 hr at 60˚C. Cells of interest were selected by observing the flat Epon embedded cell monolayer under the light microscope, and mounted on pre-polymerized Epon blocks for thin sectioning. Ultrathin sections (~90 nm) were cut parallel to the cell monolayer and collected on single-slot, formvar-coated copper grids, and stained in uranyl acetate and lead citrate. Synapses were selected at low magnification using a JEOL 1010 electron microscope. All analyses were performed on single ultrathin sections of randomly selected synapses. The distribution of DCVs was measured with ImageJ on digital images of synapses taken at 100,000× magnification using analysis software (Soft Imaging System, Gmbh, Germany). The observer was blinded for the genotype. For all morphological analyses we selected only synapses with intact synaptic plasma membranes with a recognizable pre and postsynaptic density. Docked DCVs had a distance of 0 nm from the vesicle membrane to the plasma membrane.

### Statistics

Student's *t* tests for unpaired data were used, throughout the paper, unless otherwise specified. If deviations differed significantly, *t* tests were Welch corrected. The Mann–Whitney test was used to compare two groups when one or both groups did not pass the normality test. To test more than two groups, Kruskal–Wallis, Bonferroni corrected, analysis of variance was used. Kolmogorov–Smirnov test was used to test whether distributions were normally distributed. Data are plotted as mean with standard error of the mean; n represents number of neurons, N the number of independent experiments.

### Acknowledgements

The authors would like to thank Prof. Nils Brose for CAPS-1/2 mice. They also thank R Zalm for cloning and producing viral particles, D Schut, I Saarloos and B Beuger for cell cultures, J Hoetjes and F den Oudsten for genotyping and R Dekker for electron microscopy. They are grateful to J Wortel and C van der Meer for animal breeding. Electron microscopy was performed at the VU/VUmc EM facility. This work is supported by the Netherlands Organization for Scientific Research (ZonMw-VENI 916-66-101, ZonMW-TOP 91208017 to RFT; Pionier/VICI 900-01-001 and ZonMW 903-42-095 to MV). This work is also supported by the EU (EUSynapse project 019055, EUROSPIN project HEALTH-F2-2009-241498, HEALTH-F2-2009-242167 SynSys project and ERC advanced grant 322966 to MV); MF is a recipient of an Erasmus Mundus Joint Doctorate grant (EU 2011-1632/001-001-EMJD) and CMP is supported by ERC Advanced grant 322966 to MV.

# Additional information

## Funding

| Funder | Grant reference number | Author |
|---|---|---|
| Netherlands Organisation for Health Research and Development (ZonMw) | 916-66-101 | Ruud F Toonen |
| Netherlands Organisation for Health Research and Development (ZonMw) | 91208017 | Ruud F Toonen |
| Netherlands Organisation for Health Research and Development (ZonMw) | 90342095 | Matthijs Verhage |
| Netherlands Organisation for Health Research and Development (ZonMw) | 90001001 | Matthijs Verhage |
| Education, Audiovisual and Culture Executive Agency (EACEA) | Erasmus Mundus Joint Doctorate grant EU 2011-1632/ 001-001-EMJD | Margherita Farina |
| European Research Council (ERC) | 322966 | Matthijs Verhage |
| Sixth Framework Programme | EUSynapse 019055 | Matthijs Verhage |
| Seventh Framework Programme | HEALTH-F2-2009-241498 | Matthijs Verhage |
| Seventh Framework Programme | SynSys project HEALTH-F2-2009-242167 | Matthijs Verhage |

The funders had no role in study design, data collection and interpretation, or the decision to submit the work for publication.

## Author contributions

MF, Conception and design, Acquisition of data, Analysis and interpretation of data, Drafting or revising the article; RB, Acquisition of data, Analysis and interpretation of data; EH, Performed electrophysiological recordings; CMP, Performed immunocytochemistry experiments; JRTW, Contributed EM data; JHB, Data processing and analysis; MV, Conception and design, Drafting or revising the article; RFT, Conception and design, Analysis and interpretation of data, Drafting or revising the article

## Author ORCIDs

Ruud F Toonen, http://orcid.org/0000-0002-9900-4233

## Ethics

Animal experimentation: Animals were housed, handled and bred according to institutional, Dutch and U.S. governmental guidelines. All animals were handled according to approved VU University Animal Ethics and Welfare Committee protocols (DEC-FGA-13-03 and DEC-FGA-14-01).

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
