## [Decision Letter]

Thank you for sending your work entitled “CAPS-1 promotes fusion competence of stationary dense-core vesicles in presynaptic terminals of mammalian neurons” for consideration at *eLife*. Your article has been generally favorably evaluated by Vivek Malhotra (Senior editor), a Reviewing editor, and 3 reviewers. However, several issues were identified that require major revisions before a final decision can be made.

The Reviewing editor and the reviewers discussed their comments before we reached this decision, and the Reviewing editor has assembled the following comments to help you prepare a revised submission.

In this article, the authors convincingly demonstrate that CAPS positively regulates the exocytosis of stationary large-dense core granules in presynaptic nerve terminals. They used mouse KO, cultured neurons particularly autotaptic, electrophysiology and time-lapsed imaging to reach this conclusion. The authors examine subcellular CAPS-1 localization, DCV localization, DCV loading, DCV docking and fusion in CAPS DKO mutants. The main conclusion of this paper is that CAPS is involved in priming DCVs in neurons, but the manuscript explores CAPS function in detail at many levels and touches on many controversies. There are a large number of conclusions from these data, many at odds with existing literature. The experiments in this manuscript are well designed and executed. Although a role for CAPS in DCV priming is well-established, this manuscript represents the most complete study of CAPS function in DCV secretion in neurons.

Major comments:

1) The authors attempt to distinguish what they term kiss-and-run from full fusion by employing NPY-pHluorin and NPY-mCherry. They show time course for fluorescent decrease for an event with the latter (Figure 3) but not the former. There are several mistaken notions in these studies. Full release of cargo (shown in Figure 3) implies nothing about the mode of fusion other than the fusion pore has dilated sufficiently to allow full cargo release. The fusion pore may dilate and reclose and still allow full release of NPY-mCherry (or NPY-pHluorin, not shown). Differences in numbers of events detected with each NPY fusion protein is very likely due to sensitivity of detection, looking for a decrease in red compared to a 100x increase in green fluorescence. This entire section seems to be mistakenly rationalized and inadequately documented.

2) Figure 2: What is going on in the bottom cartoon? The text says that semaphorin is secreted, but the figure indicates that there is no secretion. It also indicates that reacidification is instantaneous. Also, it is unclear why the cartoon shows a kiss and run type of event given the rapid spike-like nature of the semaphluorin fluorescence. The authors may wish to consider removing all points and discussions about kiss-and-run vs. full fusion if their available data cannot distinguish between these mechanisms.

3) There are several errors in citations. The authors cite [38], as evidence that there is no docking defect in *Drosophila* dCAPS mutants. This study did not analyze docking directly, and rather report an increase in clustering of presynaptic DCVs. Furthermore, they cite [27] (Introduction section) as a *Drosophila* study; in fact this paper assays fusion in chromaffin cells. As the authors suggest, no defect in DCV docking was observed in chromaffin cells by [27]. The authors reference [15] for demonstrating that the MUN domains of CAPS and Munc13 bind syntaxin differently. The correct reference is [36]. The [15] paper is in complete disagreement with this manuscript, nevertheless the authors should cite this manuscript for first demonstrating that open syntaxin bypasses the requirement for CAPs (paragraph two of the subsection “A post-docking role for CAPS-1 in DCVs Fusion” in the Discussion). The authors cite [26], for demonstrating that CAPS is required for docking DCVs at C*. elegans* NMJs (“CAPS/UNC-31 deletion […] between control and CAPS DKO” and “Hence, with the exception of *C. elegans unc-31* null mutants that do show docking defects […] CAPS-1 functions as a post-docking priming protein for DCV release”); in fact this manuscript describes a TIRF assay in cultured neurons, moreover the correct reference for the TIRF assay is [62]. The reference for DCV docking at *C. elegans* neuromuscular junctions in CAPS mutants is [15]. Finally, the authors should cite [19], which demonstrates that there is no defect in docking DCVs in CAPS DKOs by electron tomography, which is a more sensitive assay for docking than TEM.

4) Cargo loading (mentioned in the subsection “DCV secretion is severely reduced upon CAPS deletion in hippocampal neurons”): The authors claim that they demonstrate that loading of cargo is not affected in DCVs in hippocampal neurons in conflict from the results of [52]). Speidel et al. were looking at catecholamine loading in chromaffin cells; these authors did not claim that there were defects in loading peptide cargo, and the current results are not incompatible with these results. The authors should make this clear. Nevertheless, it is worth stating clearly which results support or contradict existing literature.

5) Abstract: What is “synaptic retention of CAPS”? Maybe “localization of CAPS at synapses” would be better.

6) Figure 2: Is “field of view” a legitimate denominator, given that the objective could be parked over a region devoid of axons? That should be dealt with in the Methods, “Fields were selected for presence of labeled sema-positive cells”. A more informative Y-axis would note that these were stimulated samples, “fusions/24 seconds stimulation”. The pause interval should be noted in the legend “16 bursts 50AP at 50Hz every 0.5 seconds”.

7) In CAPS DKO release occurs “only after the second burst of stimulation” and “delaying the onset of DCV release upon stimulation.” It appears from the curves to be half the rate for the CAPS DKO at all time points, even within the first burst.

8) Figure 4: A figure with larger electron micrographs should be included to show context. Additionally, an enlargement showing what the authors score as docked and close but undocked DCVs should be included.

9) In the Discussion the authors discuss the different roles of CAPS and Munc13 for the release of DCVs at extrasynaptic versus synaptic sites. In addition there seems to be a third component mediating DCV fusion, since some release at extrasynaptic sites and secretion of mobile vesicles remains in CAPS mutants. If that is correct, it is worth stating in the Discussion.

10) “DCV secretion was reduced by 70% in CAPS-1/CAPS-2 DKO neurons and remaining fusion events required prolonged stimulation”. This was stated in the Abstract and at several points in the manuscript corresponding to Figures 2 and 3. Any differences between control and CAPS DKO time courses of cumulative events were not at all evident to this reviewer.

11) “CAPS deletion specifically reduced secretion of stationary DCVs”. This was stated in the Abstract and at several points in the manuscript corresponding to Figure 4. Whereas fusion events for stationary DCVs were substantially decreased in DKO, the number of events for moving DCVs is very small and it was not clear that a decrease would be observable above background.

12) “… but CAPS-1-EYFP and DCVs rarely traveled together” (in the Abstract). This is only anecdotally documented. In fact, the text noted such an occurrence that contrasts with this statement. Because the authors suggest that some CAPS-1 associates with some DCVs (see below), this becomes an important point but it is not documented.

13) “Synaptic retention of CAPS-1-EYFP in DKO neurons was calcium-dependent…” (in the Abstract). This corresponds to Figure 6, which is the most interesting observation in this study. However, it shows a stimulation-dependent decrease of CAPS-1-EYFP from synapses. So the loss of CAPS-1-EYFP from synapses, not its retention, was calcium-dependent or calcium-stimulated. While the authors comment on this observations from a physiological perspective of adaptation, there is no suggestion about underlying mechanism for CAPS-1 or other proteins in which similar calcium-dependent synaptic loss has been observed.

14) “Deletion of CAPS-1 and CAPS-2 did not affect DCV… docking…”. Docking corresponds to a putative functional state of DCVs. It has been measured by conventional fixation EM, rapid freeze EM, and TIRF microscopy for which there is frequent disagreement. The authors used the first of these techniques and note in the text that their results (lack of effect of CAPS KO on docking) differ from results in *C. elegans*. Importantly, the *C. elegans* results utilized rapid freeze EM. Because of known shrinkage and fixation artifacts in conventional EM, the authors should refrain from using the term docking but describe their results as DCVs close to the membrane by EM. It may well be the case when re-examined using other techniques, that there are effects of CAPS deletion on DCV docking as has been shown for Munc13-1.

15) In the single cell studies, the authors use the CAPS2 KO as controls to compare with CAPS DKO cells. While they cite [22], as the basis for this choice, the Jockusch et al. study did not view DCV exocytosis so the rationale for this is unclear. Labeling the figures as “control” provides confusion and should be labeled CAPS2 KO. The authors should compare cells in their assay from all genotypes to rationalize this choice.

[Editors' note: further revisions were requested prior to acceptance, as described below.]

Thank you for resubmitting your work entitled “CAPS-1 promotes fusion competence of stationary dense-core vesicles in presynaptic terminals of mammalian neurons” for further consideration at *eLife*. Your revised article has been favorably evaluated by Vivek Malhotra (Senior editor), a Reviewing editor, and three reviewers. The manuscript has been improved but there are some remaining issues that need to be addressed before acceptance, as outlined below:

*Comments by reviewer 1*:

1) Figure 2: In the rebuttal (point #2) the authors state that they are showing NPY-pHluorin in Figure 2, but they probably mean semaphorin-pHluorin.

2) In Figure 2, the plasma membrane for the docked vesicle on the left is broken. What happened to the membrane between the TM domains of syntaxin? Is this meant to illustrate a hemi-fusion?

3) The cartoon shows a vesicle opening and the contents rushing from it. In the bottom row, three events are shown. One appears to be a full fusion that does not result in reacidification of the cargo. Two more appear to indicate that the vesicle reacidified without any release of cargo. These seem to have no relationship with the cartoon. There is no explanation of these two kinds of events in the figure legend or in the text (Results section).

4) In the legend it says that this sudden increase in fluorescence is counted as a fusion event in panels C and E. Does that mean only the one with a permanent increase was scored as fusion events? Or were any increases counted as fusion events? Should these be labeled as “complete release” or “incomplete release”, as described for NPY in the rebuttal? The legend and text needs to be less ambiguous, for example: “Both complete and incomplete release of Semaphorin as labeled in Figure 3 were counted as fusion events in panels C and E.”

5) Figure 3. The different assays scored in the top row versus second row in Figure 3 are not clear from looking at the figure. The authors should label the top row 'NPY-pHluorin' and the second row 'NPY-mCherry'.

6) The events in Figure 3 are all fusions, regardless of whether they were complete or incomplete release. Can the authors indicate what the complete and incomplete fusions look like for NPY-pHluorin. The authors indicate what NPY-mCherry complete release events look like In Figure 3, but there is no indication what the events in the top row might look like. This is the exact same problem as in Figure 2. It is just not clear what is being scored and how these events are assembled into the panels in the figure.

7) Figure 3 is labeled 'fusion events', and Figure 3 is labeled 'events'. One assumes from looking at the figure that they are the same events. Using 'fusions / cell' and 'complete release / cell' would be clearer.

8) Figure 3 shows that 50% of the DKO cells show no release. The authors should make it clear that these cells were excluded from the analyses showing the 50% decrease in the rate of fusions in the top rows. (Otherwise the data can be explained simply by the fact that a fraction of cells failed to respond, not that their rates of fusion were affected).

*Comments by reviewer 3*:

9) There remains some confusion about studies in Figure 3 and their re-interpretation. This issue can be considered minor but some further clarification would be useful. In the original version of the manuscript, increased fluorescence of a pHluorin construct (Figure 3) versus complete loss of fluorescence of an mCherry construct (Figure 3) were interpreted to indicate fusion pore opening and full fusion, respectively. The revised manuscript more appropriately describes what these data indicate but it remains unclear whether these data support the interim conclusion that “incomplete and complete cargo release events were affected (by CAPS KO) to the same extent”. It depends on how these experiments were done. If they were done as suggested in the rebuttal letter: “we compared fusion pore opening (step-like increases in the green channel) to full cargo release (complete loss of signal in the red channel) and concluded that fusion pore openings occur more often than full cargo release” using both channels to detect single events, then the authors' conclusion would stand because they would show pHluorin brightening events that were accompanied by complete or incomplete mCherry loss. If this is the case, the figure legend should indicate this and the authors could consider showing traces pHluorin brightening that were accompanied by either complete or incomplete mCherry loss. If, however, the number of pHluorin brightening’s and mCherry complete losses were determined independently, then it could be the case that sensitivity of event detection favors pHluorin brightening events. From what is currently described in the manuscript, it is unclear.

10) In the revised manuscript, there is an expanded discussion of how CAPS functions with an emphasis on its PH domain and PIP2. This section needs editing and greater attention to the references cited (see the Nguyen Truong paper for references). What is PLC16? See Kabachinski et al. for discussion of PLCs in CAPS function.

---

## [Author Response]

We thank the reviewers and the editor for their insightful comments and constructive suggestions. We have made changes to our manuscript to fully comply with the reviewers’ suggestions and added new data (Figures 1, 2, 4 and 7; and Figure 4–figure supplement 1) and better analysis and discussion of the data, as requested. Together, we feel this has considerably strengthened the conclusions of the data.

*Major comments*:

*1) The authors attempt to distinguish what they term kiss-and-run from full fusion by employing NPY-pHluorin and NPY-mCherry. They show time course for fluorescent decrease for an event with the latter (*Figure 3*) but not the former. There are several mistaken notions in these studies. Full release of cargo (shown in*
Figure 3*) implies nothing about the mode of fusion other than the fusion pore has dilated sufficiently to allow full cargo release. The fusion pore may dilate and reclose and still allow full release of NPY-mCherry (or NPY-pHluorin, not shown). Differences in numbers of events detected with each NPY fusion protein is very likely due to sensitivity of detection, looking for a decrease in red compared to a 100x increase in green fluorescence. This entire section seems to be mistakenly rationalized and inadequately documented*.

The reviewers argue that our interpretation of the data in terms of vesicle ‘fusion modes’ is not justified and that partial release might not be detected in the red channel due to sensitivity issues. We agree on both these issues. We have now removed interpretation in terms of fusion modes and made the text more factual. It is not the aim of this study to interpret fusion modes and this should not interfere with the main message on CAPS function. Throughout the manuscript we changed the text: completely disappearing NPY-mCherry puncta (Figure 3) represent “complete cargo release” (avoiding the term ‘full fusion’), and NPY-pHluorin events (Figure 3) represent fusion pore opening (avoiding the term ‘kiss and run’) with either “complete release” or “incomplete release”, depending on loss in the red channel or not. Regarding the sensitivity issues: we compared fusion pore opening (step-like increases in the green channel) to *full* cargo release (complete loss of signal in the red channel) and concluded that fusion pore openings occur more often than full cargo release. We changed the text (” DCV fusion pore opening can progress to complete release […] synaptic preference of DCV release.”) to make this clearer.

*2)*
Figure 2*: What is going on in the bottom cartoon? The text says that semaphorin is secreted, but the figure indicates that there is no secretion. It also indicates that reacidification is instantaneous. Also, it is unclear why the cartoon shows a kiss and run type of event given the rapid spike-like nature of the semaphluorin fluorescence. The authors may wish to consider removing all points and discussions about kiss-and-run vs. full fusion if their available data cannot distinguish between these mechanisms*.

We changed the cartoon to better represent the data, as suggested by the reviewers. For our analysis of fusion events in Figure 2 we used the sudden increase in NPY-pHluorin fluorescence as signal for fusion pore opening. We added text in figure legend and results to better explain this. In line with comment #1 we removed our comments on vesicle fusion modes.

*3) There are several errors in citations. The authors cite*
[38]*, as evidence that there is no docking defect in* Drosophila *dCAPS mutants. This study did not analyze docking directly, and rather report an increase in clustering of presynaptic DCVs. Furthermore, they cite*
[27]
*(Introduction section) as a* Drosophila *study; in fact this paper assays fusion in chromaffin cells. As the authors suggest, no defect in DCV docking was observed in chromaffin cells by*
[27]*. The authors reference*
[15]
*for demonstrating that the MUN domains of CAPS and Munc13 bind syntaxin differently. The correct reference is*
[36]*. The*
[15]
*paper is in complete disagreement with this manuscript, nevertheless the authors should cite this manuscript for first demonstrating that open syntaxin bypasses the requirement for CAPs (paragraph two of the subsection “A post-docking role for CAPS-1 in DCVs Fusion” in the Discussion). The authors cite*
[26]*, for demonstrating that CAPS is required for docking DCVs at* C. elegans *NMJs (“CAPS/UNC-31 deletion […] between control and CAPS DKO” and “Hence, with the exception of* C. elegans *unc-31 null mutants that do show docking defects […] CAPS-1 functions as a post-docking priming protein for DCV release”); in fact this manuscript describes a TIRF assay in cultured neurons, moreover the correct reference for the TIRF assay is*
[62]*. The reference for DCV docking at* C. elegans *neuromuscular junctions in CAPS mutants is*
[15]*. Finally, the authors should cite*
[19]*, which demonstrates that there is no defect in docking DCVs in CAPS DKOs by electron tomography, which is a more sensitive assay for docking than TEM.*

The reviewers identified several errors in citations. We fully agree and apologize for these mistakes. We corrected them and added [19], to the discussion on DCV docking.

*4) Cargo loading (mentioned in the subsection “DCV secretion is severely reduced upon CAPS deletion in hippocampal neurons”): The authors claim that they demonstrate that loading of cargo is not affected in DCVs in hippocampal neurons in conflict from the results of*
[52]*). Speidel et al. were looking at catecholamine loading in chromaffin cells; these authors did not claim that there were defects in loading peptide cargo, and the current results are not incompatible with these results. The authors should make this clear. Nevertheless, it is worth stating clearly which results support or contradict existing literature*.

The reviewers identified an ambiguous statement about cargo loading. The reviewers are correct. We changed the sentence to better describe that *protein* loading is not affected in hippocampal neurons whereas *catecholamine* loading in CAPS-1 KO chromaffin cells is.

*5) Abstract: What is* “*synaptic retention of CAPS*”*? Maybe* “*localization of CAPS at synapses*” *would be better*.

We changed “synaptic retention of CAPS” to “synaptic localization of CAPS ” as suggested.

*6)*
Figure 2*: Is* “*field of view*” *a legitimate denominator, given that the objective could be parked over a region devoid of axons? That should be dealt with in the Methods,* “*Fields were selected for presence of labeled sema-positive cells*”*. A more informative Y-axis would note that these were stimulated samples,* “*fusions/24 seconds stimulation*”*. The pause interval should be noted in the legend* “*16 bursts 50AP at 50Hz every 0.5 seconds*”.

The reviewers question whether “field of view” is a legitimate denominator (Figure 2). We added a better description in the Methods section and changed y-axis labels in Figure 2 to “fusion events/per field of view” and added “16 bursts 50AP at 50Hz every 0.5 seconds” to the legend and Methods section, as suggested.

*7) In CAPS DKO release occurs* “*only after the second burst of stimulation*” *and* “*delaying the onset of DCV release upon stimulation.*” *It appears from the curves to be half the rate for the CAPS DKO at all time points, even within the first burst*.

The reviewers question our conclusion that release onset is delayed in CAPS DKO. We re-analysed the time to fusion in CAPS DKO vs controls and did not find statistically significant differences and removed this conclusion in all parts of the manuscript.

*8)*
Figure 4*: A figure with larger electron micrographs should be included to show context. Additionally, an enlargement showing what the authors score as docked and close but undocked DCVs should be included*.

We have included new figures with larger electron micrographs, as requested by the reviewers (Figure 4) and an enlargement showing how docked and close but undocked DCVs were scored (Figure 4–figure supplement 1).

*9) In the Discussion the authors discuss the different roles of CAPS and Munc13 for the release of DCVs at extrasynaptic versus synaptic sites. In addition there seems to be a third component mediating DCV fusion, since some release at extrasynaptic sites and secretion of mobile vesicles remains in CAPS mutants. If that is correct, it is worth stating in the Discussion*.

The reviewers point out that an additional third component may mediate extrasynaptic release as both in Munc13 and CAPS DKOs some fusion remains. We agree and discuss this possibility in the Discussion on the subsection headed “A post-docking role for CAPS-1 in DCVs fusion”.

*10)* “*DCV secretion was reduced by 70% in CAPS-1/CAPS-2 double null mutant (DKO) neurons and remaining fusion events required prolonged stimulation*”*. This was stated in the Abstract and at several points in the manuscript corresponding to*
Figures 2 and 3*. Any differences between control and CAPS DKO time courses of cumulative events were not at all evident to this reviewer*.

This point is in fact the same issue as addressed at point 7. We agree and removed these statements throughout the manuscript.

*11)* “*CAPS deletion specifically reduced secretion of stationary DCVs*”*. This was stated in the Abstract and at several points in the manuscript corresponding to*
Figure 4*. Whereas fusion events for stationary DCVs were substantially decreased in DKO, the number of events for moving DCVs is very small and it was not clear that a decrease would be observable above background*.

The reviewers argue that fusion events of mobile vesicles are rare and effects on CAPS loss might therefore remain unnoticed. We changed the text to “CAPS deletion strongly reduced secretion of stationary DCVs” and removed the term “specifically” throughout the manuscript.

*12)* “*… but CAPS-1-EYFP and DCVs rarely traveled together*” *(in the Abstract). This is only anecdotally documented. In fact, the text noted such an occurrence that contrasts with this statement. Because the authors suggest that some CAPS-1 associates with some DCVs (see below), this becomes an important point but it is not documented*.

The reviewers state that documentation of CAPS-1-EYFP and DCV co-trafficking is too anecdotal and deserves better documentation. We agree and added a new, quantitative analysis of DCV and CAPS-1-EYFP dynamics to Figure 7 (new Figure 7) to support our conclusion that co-trafficking of CAPS-1 with mobile DCVs is very rare.

*13)* “*Synaptic retention of CAPS-1-EYFP in DKO neurons was calcium-dependent…*” *(in the Abstract). This corresponds to*
Figure 6*, which is the most interesting observation in this study. However, it shows a stimulation-dependent decrease of CAPS-1-EYFP from synapses. So the loss of CAPS-1-EYFP from synapses, not its retention, was calcium-dependent or calcium-stimulated. While the authors comment on this observations from a physiological perspective of adaptation, there is no suggestion about underlying mechanism for CAPS-1 or other proteins in which similar calcium-dependent synaptic loss has been observed*.

The reviewers argue that our data in Figure 6 argue for Ca^2+^-dependence of synaptic dispersion, not retention and notes that there are no suggestions about underlying mechanisms. We agree and changed the text to “Synaptic dispersion of CAPS-1-EYFP in DKO neurons was calcium-dependent”. We have recently shown that synaptic dispersion and reclustering of Munc18-1 allows neurons to modulate synaptic strength upon network activity (8). Reclustering of Munc18-1 required PKC activation and was syntaxin 1 independent. CAPS1 membrane association requires its PIP2-binding pleckstrin homology (PH) domain. Robust Ca2+ influx in our experiments likely activates phospholipase C (PLC) which hydrolysis PIP2 and may trigger CAPS1 dispersion from synapses. Recent work on CAPS2 shows that the PH-domain is required to prime vesicles (35). Calcium-dependent PIP2 hydrolysis may therefore act as negative feedback mechanism reducing CAPS1 availability at the synapse upon robust activity. In addition to direct membrane interaction, CAPS1 also binds syntaxin via its MUN domain. Syntaxin also disperses from synapses, which may result in CAPS1 co-dispersion. We added this to the Discussion ” This is also an attractive explanation[…]after high frequency stimulation”.

*14)* “*Deletion of CAPS-1 and CAPS-2 did not affect DCV… docking…*”*. Docking corresponds to a putative functional state of DCVs. It has been measured by conventional fixation EM, rapid freeze EM, and TIRF microscopy for which there is frequent disagreement. The authors used the first of these techniques and note in the text that their results (lack of effect of CAPS KO on docking) differ from results in* C. elegans*. Importantly, the* C. elegans *results utilized rapid freeze EM. Because of known shrinkage and fixation artifacts in conventional EM, the authors should refrain from using the term docking but describe their results as DCVs close to the membrane by EM. It may well be the case when re-examined using other techniques, that there are effects of CAPS deletion on DCV docking as has been shown for Munc13-1*.

The reviewers argue that chemical fixation used in our study might produce artifacts that change the precise distance between vesicles and the membrane. As a consequence, vesicles touching the membrane in fixed tissue may in fact have been at a short distance from the membrane. This is true and we have considered it common knowledge, but we have now added a statement on this in the text. We feel that this fact does not preclude the use of the term ‘docking’ and that the suggested alternative ‘DCVs close to the membrane’ is not better, because this might give the impression that DCVs are positioned accidentally near a membrane. The term ‘docking’, like many words in shipping, has a Dutch origin, in this case the word ‘dok’, which describes a situation in which a vessel can be loaded or unloaded. Whether or not the vessel touches the quay is not defined and typically there is some space between vessel and quay. The essential thing is the functional state (to permit (un)loading), very similar to the reviewer’s definition for vesicles in synapses (‘docking corresponds to a putative functional state’).

Although earlier reports using chemical fixation of mouse synapses did not find differences in DCV localization (22). The reviewers are right that future studies using cryo-fixation might unmask (subtle) docking defects upon CAPS loss. In fact, Imig et al., using cryo-fixation, found a reduction in the number of docked SVs and a lower, albeit not statistically significant number of DCVs within 200 nm of the active zone between CAPS DKO and controls. We added text to the Discussion to point this out.

*15) In the single cell studies, the authors use the CAPS2 KO as controls to compare with CAPS DKO cells. While they cite*
[22]*, as the basis for this choice, the Jockusch et al. study did not view DCV exocytosis so the rationale for this is unclear. Labeling the figures as* “*control*” *provides confusion and should be labeled CAPS2 KO. The authors should compare cells in their assay from all genotypes to rationalize this choice*.

We agree with the reviewers that although synaptic transmission is not affected in CAPS 2 KO neurons, DCV release might. We can indeed not exclude this and we therefore might underestimate the effect of CAPS DKO in Figure 3. However, we used wild-type controls in Figure 2 and CAPS DKO leads to a 70% reduction of release events compared to either control. We renamed “control” to “CAPS 2KO” to avoid confusion.

[Editors' note: further revisions were requested prior to acceptance, as described below.]

We have changed Figures 2 and 3 as suggested by reviewer 1 and added Figure 2–figure supplement 2 to better explain the different fusion modes reported by Semaphluorin. Please find below the point-by-point answers to the reviewers’ comments.

Comments by reviewer 1:

*1)*
Figure 2*: In the rebuttal (point #2) the authors state that they are showing NPY-pHluorin in*
Figure 2*, but they probably mean semaphorin-pHluorin*.

The reviewer is correct and we are sorry for this mistake, Figure 2 indeed shows semaphorin-pHluorin.

*2) In*
Figure 2*, the plasma membrane for the docked vesicle on the left is broken. What happened to the membrane between the TM domains of syntaxin? Is this meant to illustrate a hemi-fusion*?

No, this appears to be caused by conversion between Illustrator versions, we changed this so that the membrane is continuous in Figure 2. We thank the reviewer for pointing this out.

*3) The cartoon shows a vesicle opening and the contents rushing from it. In the bottom row, three events are shown. One appears to be a full fusion that does not result in reacidification of the cargo. Two more appear to indicate that the vesicle reacidified without any release of cargo. These seem to have no relationship with the cartoon. There is no explanation of these two kinds of events in the figure legend or in the text (Results section)*.

The reviewer is correct; the cartoon does not represent the different fusion modes reported by Semaphluorin. The cartoon and fluorescence traces in Figure 2 illustrate the definition of fusion events counted in Figure 2: a sudden increase in fluorescence two standard deviations above initial fluorescence (F0). Hence, all 3 events were counted as fusion events. This was not clear in the original figure and we changed the figure adding a grey dotted line representing the 2xSD, and added text to the figure legend and main text (Results section) to make this clearer. We realize that the different traces may raise confusion. Therefore, we added Figure 2–figure supplement 2 to better explain the different fusion modes reported by SemapHluorin.

*4) In the legend it says that this sudden increase in fluorescence is counted as a fusion event in panels C and E. Does that mean only the one with a permanent increase was scored as fusion events? Or were any increases counted as fusion events? Should these be labeled as* “*complete release*” *or* “*incomplete release*”*, as described for NPY in the rebuttal? The legend and text needs to be less ambiguous, for example:* “*Both complete and incomplete release of Semaphorin as labeled in*
Figure 3
*were counted as fusion events in panels C and E*.”

We are sorry for the confusion; all 3 events are fusion events. In Figure 2, we counted all events with an increase in fluorescence two standard deviations above F0. The onset of fusion was defined as the first frame with an increase two times SD above F0 (added to the Materials and methods section). In line with comment #3 we changed Figure legend and text (Results section) to make this clearer.

*5)*
Figure 3*. The different assays scored in the top row versus second row in*
Figure 3
*are not clear from looking at the figure. The authors should label the top row 'NPY-pHluorin' and the second row 'NPY-mCherry'*.

We changed the figure as suggested.

*6) The events in*
Figure 3
*are all fusions, regardless of whether they were complete or incomplete release. Can the authors indicate what the complete and incomplete fusions look like for NPY-pHluorin. The authors indicate what NPY-mCherry complete release events look like In*
Figure 3*, but there is no indication what the events in the top row might look like. This is the exact same problem as in*
Figure 2*. It is just not clear what is being scored and how these events are assembled into the panels in the figure*.

Similar to our analysis of fusion events in Figure 2, in Figure 3 all sudden increases of fluorescence two standard deviations above F0 were counted as fusion events. We added this to figure legend and text. We also added a typical NPY-pHluorin trace to Figure 3, which reports a fusion event counted in 3B. After the increase in fluorescence, the decline in fluorescence may represent either NPY-Phluorin cargo diffusion (‘complete release’) or vesicle resealing and re-acidification (‘incomplete release’). In Figure 3 we did not discriminate between the two and scored all events > 2xSD above F0.

*7)*
Figure 3
*is labeled 'fusion events', and*
Figure 3
*is labeled 'events'. One assumes from looking at the figure that they are the same events. Using 'fusions / cell' and 'complete release / cell' would be clearer*.

We agree and changed the figure as suggested.

*8)*
Figure 3
*shows that 50% of the DKO cells show no release. The authors should make it clear that these cells were excluded from the analyses showing the 50% decrease in the rate of fusions in the top rows. (Otherwise the data can be explained simply by the fact that a fraction of cells failed to respond, not that their rates of fusion were affected)*.

We fully agree and added this statement to the figure legend and main text. We would like to thank reviewer 1 for comments and suggestions. They significantly improved the legibility of our manuscript.

Comments by reviewer 3:

*9) There remains some confusion about studies in*
Figure 3
*and their re-interpretation. This issue can be considered minor but some further clarification would be useful. In the original version of the manuscript, increased fluorescence of a pHluorin construct (*Figure 3*) versus complete loss of fluorescence of an mCherry construct (*Figure 3*) were interpreted to indicate fusion pore opening and full fusion, respectively. The revised manuscript more appropriately describes what these data indicate but it remains unclear whether these data support the interim conclusion that* “*incomplete and complete cargo release events were affected (by CAPS KO) to the same extent*”*. It depends on how these experiments were done. If they were done as suggested in the rebuttal letter:* “*we compared fusion pore opening (step-like increases in the green channel) to full cargo release (complete loss of signal in the red channel) and concluded that fusion pore openings occur more often than full cargo release*” *using both channels to detect single events, then the authors' conclusion would stand because they would show pHluorin brightening events that were accompanied by complete or incomplete mCherry loss. If this is the case, the figure legend should indicate this and the authors could consider showing traces pHluorin brightening that were accompanied by either complete or incomplete mCherry loss. If, however, the number of pHluorin brightening’s and mCherry complete losses were determined independently, then it could be the case that sensitivity of event detection favors pHluorin brightening events. From what is currently described in the manuscript, it is unclear*.

In Figure 3 the number of pHluorin brightening’s and mCherry complete losses were determined independently and the reviewer is correct that we therefore cannot formally exclude that sensitivity of event detection favors pHluorin brightening events. Which would explain why we see more pHluorin events than mCherry disappearances. To exclude any misinterpretation, we changed the concluding sentence of this paragraph to “Thus, deletion of CAPS-1 resulted in a strong reduction of DCV fusion events reported by NPY-Phluorin or NPY-mCherry”.

*10) In the revised manuscript, there is an expanded discussion of how CAPS functions with an emphasis on its PH domain and PIP2. This section needs editing and greater attention to the references cited (see the Nguyen Truong paper for references). What is PLC16? See Kabachinski et al. for discussion of PLCs in CAPS function*.

We thank the reviewer for pointing this out. We apologize and corrected citations as suggested. PLC16 is a typo and should state PLC.